# Climate signals in a multispecies tree-ring network from central and southern Italy and reconstruction of the late summer temperatures since the early 1700s

Giovanni Leonelli[1], Anna Coppola[2], Maria Cristina Salvatore[2], Carlo Baroni[2,3], Giovanna Battipaglia[4,5], Tiziana Gentilesca[6], Francesco Ripullone[6], Marco Borghetti[6], Emanuele Conte[7], Roberto Tognetti[7], Marco Marchetti[7], Fabio Lombardi[8], Michele Brunetti[9], Maurizio Maugeri[9,10], Manuela Pelfini[11], Paolo Cherubini[12], Antonello Provenzale[3], Valter Maggi[1,3]

[1] Università degli Studi di Milano–Bicocca — Dept. of Earth and Environmental Science
[2] Università degli Studi di Pisa — Dept. of Earth Science
[3] Istituto di Geoscienze e Georisorse, Consiglio Nazionale delle Ricerche, Pisa
[4] Università della Campania — Dept. DiSTABiF
[5] University of Montpellier 2 (France) — PALECO EPHE
[6] Università degli Studi della Basilicata — School of Agricultural, Forestry, Food and Environmental Sciences, Potenza
[7] Università degli Studi del Molise — Dept. of Bioscience and Territory
[8] Università Mediterranea di Reggio Calabria — Dept. of Agronomy
[9] Istituto di Scienze dell'Atmosfera e del Clima, Consiglio Nazionale delle Ricerche, Bologna
[10] Università degli Studi di Milano — Dept. of Physics
[11] Università degli Studi di Milano — Dept. of Earth Science
[12] Swiss Federal Institute for Forest, Snow and Landscape Research WSL (Switzerland)

*Correspondence to*: Giovanni Leonelli (*giovanni.leonelli@unimib.it*)

**Abstract.** A first assessment of the main climatic drivers that modulate the tree-ring width (RW) and maximum latewood density (MXD) along the Italian Peninsula and north-eastern Sicily was performed using 27 forest sites, which include conifers (RW and MXD) and broadleaves (only RW). Tree-ring data were compared using the correlation analysis of the monthly and seasonal variables of temperature, precipitation and standardized precipitation index (SPI, used to characterize meteorological droughts) against each species-specific site chronology and against the highly sensitive to climate (HSTC) chronologies (based on selected indexed individual series). We find that climate signals in conifer MXD are stronger and more stable over time than those in conifer and broadleaf RW. In particular, conifer MXD variability is directly influenced by the late summer (August, September) temperature and is inversely influenced by the summer precipitation and droughts (SPI at a timescale of 3 months). The MXD sensitivity to AS temperature and to summer drought is mainly driven by the latitudinal gradient of summer precipitation amounts, with sites in the northern Apennines showing stronger climate signals than sites in the south. Conifer RW is influenced by the temperature and drought of the previous summer, whereas broadleaf RW is more influenced by summer precipitation and drought of the current growing season. The reconstruction of the late summer temperatures for the Italian Peninsula for the past 300 yr, based on the HSTC chronology of conifer MXD, shows a stable model performance that underlines periods of climatic cooling (and likely also wetter conditions) in 1699, 1740, 1814, 1914, 1938 and well follows the variability of the instrumental record and of other tree-ring based reconstructions in the region. Considering a 20 yr low-pass filtered series, the reconstructed temperature record consistently deviates <1°C from the instrumental record. This divergence may be due also to the precipitation patterns and drought stresses that influence the tree-ring MXD at our study sites. The reconstructed late-summer temperature variability is also linked to summer drought conditions and it is valid for the west-east oriented region including Sardinia, Sicily, the Italian Peninsula and the western Balkan area along the Adriatic coast.

## 1 Introduction

Reconstructions of climate for periods before instrumental records rely on proxy data from natural archives and on the ability

to date them. Among the available proxies, tree rings are one of the most used archives for reconstructing past climates with annual resolution in continental areas and they often come from the temperature-limited environments with high latitudes and altitudes (e.g., Briffa et al., 2004; Rutherford et al., 2005). Tree-ring data can be used at regional to global scales (IPCC, 2013) and long chronologies covering millennia, going back as far as the early Holocene, are available (for Europe: Becker, 1993; Friedrich et al., 2004; Nicolussi et al., 2009).

The reconstruction of past climate variability and the analysis of its effects on forest ecosystems are crucial elements for understanding climatic processes and for predicting what responses should be expected in ecosystems under the ongoing climatic and global changes. In particular, the Mediterranean region is a prominent climate change hot spot (Giorgi, 2006; Turco et al., 2015), and by the end of this century, it will likely experience a regional warming higher than the global mean (up to +5 °C in summer) and a reduction of the average summer precipitation (up to -30 %; Somot et al., 2007; IPCC, 2013). As a consequence of the poleward expansion of the subtropical dry zones (e.g., Fu et al., 2006), subtropical environments under climate change are already facing strong hydroclimatic changes due to lower precipitation and human exploitation (e.g., in southwestern north America, Seager et al., 2007; Seager and Vecchi, 2010). Moreover, in these environments (including also the Mediterranean region), soil moisture will likely drop, resulting in a contraction of temperate drylands by approximately a third (converting into subtropical drylands), and longer periods of drought in deep soil layers are expected (Schlaepfer et al., 2017). The increase in droughts conditions during the growing season is already negatively impacting tree growth, especially at xeric sites in the southwestern and eastern Mediterranean (e.g., Galván et al., 2014). At the ecosystem level, in the near future, the responses to climate changes will impact the various forest species in a different way, depending on their physiological ability to acclimate and adapt to the new environmental conditions (e.g., Battipaglia et al., 2009; Ripullone et al., 2009), and on their capacity to grow, accumulate biomass, and contribute as sinks in the terrestrial carbon cycle. Natural summer fires in the Mediterranean area are also expected to increase in frequency over the coming decades as a response to increasingly frequent drought conditions, assuming a lack of additional fire management and prevention measures (Turco et al., 2017).

## 1.1 Tree-ring Response to Climate

Climate-growth relationships have been studied for several species in the Mediterranean region, with different objectives: forest productivity (e.g., Biondi, 1999; Boisvenue and Running, 2006; Nicault et al., 2008; Piovesan et al., 2008; Babst et al., 2013), tree ecophysiology, wood formation and related dating issues (Cherubini et al., 2003; Battipaglia et al., 2014), sustainability of forest management (e.g., Boydak and Dogru, 1997; Barbati et al., 2007; Marchetti et al., 2010; Castagneri et al., 2014), provision of ecosystem services (e.g., Schröter et al., 2005) such as carbon sequestration (e.g., Scarascia-Mugnozza and Matteucci, 2014; Calfapietra et al., 2015; Borghetti et al., 2017), effective biodiversity conservation (e.g., Todaro et al., 2007; Battipaglia et al., 2009), and climate reconstruction (see next heading), which has led to a variety of associations between climate variables and growth responses in conifers and broadleaves from different environments and ecosystems. Mainly considering the species of this study, we report the main findings on the climate-growth responses found in this region.

— *Conifers*. Studies on silver fir (*Abies alba* Mill.) growth in the Italian Peninsula reveal high sensitivity to the climate of the previous summer, $August_{-1}$ in particular, positive correlations with precipitation and negative correlations with temperature (Carrer et al., 2010; Rita et al., 2014). Moreover, tree growth in this region is moderately negatively correlated with the temperature of the current summer (unlike that in stands located in the European Alps; Carrer et al., 2010), namely, high temperatures in July and August negatively affect tree growth. A dendroclimatic network of pines (*Pinus nigra* J.F. Arnold and *P. sylvestris* L.) in east-central Spain shows that drought (namely, the Standardized Precipitation-Evapotranspiration Index - SPEI; Vicente-Serrano et al., 2010) is the main climatic driver of tree-ring growth (Martin-Benito et al., 2013). In a *P. uncinata* network from the Pyrenees, an increasing influence of summer droughts (SPEI) on tree-ring

widths (RW) during the 20th century and the control of May temperatures on maximum latewood density (MXD) is found (Galván et al., 2015). However, in the above-mentioned analyses, the possible influences of the summer climate variables from the year prior to the growth were not considered. Elevation, and particularly the related moisture regime, in the eastern Mediterranean region is the main driver of tree-ring growth patterns in a multispecies conifer network comprised of *P. nigra*, *P. sylvestris* and *P. pinea* L. specimens (Touchan et al., 2016). A dipole pattern in tree-ring growth variability is reported for Mediterranean pines ranging from Spain to Turkey, with a higher sensitivity to summer drought in the East than in the West, and with higher sensitivity to early summer temperature in the West (Seim et al., 2015). A strong correlation between autumn-to-summer precipitation and between summer drought and tree-ring growth is reported for sites (mainly of conifers) in northern Africa-western Mediterranean, with trees from Morocco also responding to the North Atlantic Oscillation Index (Touchan et al., 2017).

— *Broadleaves*. In the western Mediterranean (northern Morocco, Algeria, Tunisia, Italy and southern France), deciduous oaks, including *Quercus robur* L., reveal a direct response of tree-ring growth to summer precipitation and an inverse response to summer temperature (Tessier et al., 1994). Beech (*Fagus sylvatica* L.) is particularly sensitive to soil moisture and air humidity; in past decades, long-term drought conditions have been shown to be the main factor causing a growth decline in the old-growth stands in the Apennines (Piovesan et al., 2008). Beech shows different responses to climate at high- vs. low-altitude sites (Piovesan et al., 2005), with these latter being positively affected by high May temperatures. Despite an expected higher drought sensitivity stress close to the southern limit of the distribution area, a late twentieth century tree-ring growth increase in beech has been reported in Albania (Tegel et al., 2014), thus underlining the different climate-growth responses in the Mediterranean region. Beech, indeed, presents complex climate growth-responses and also appears to be a less responsive species in the Mediterranean area when compared to conifers such as *P. sylvestris, P. nigra, P. uncinata* or *A. alba* (as found in south-east France; Lebourgeois et al., 2012).

## 1.2 Tree-ring Based Climate Reconstructions

One of the most powerful tools in terrestrial paleoclimatology is obtaining dated information about the past climate and past environmental conditions in a region by analyzing the tree rings. However, in the Mediterranean region, the low temporal stability of the recorded climatic signals (e.g., Lebourgeois et al., 2012; Castagneri et al., 2014), the scarcity of long chronologies, and the high variability of climatic and ecological conditions (Cherubini et al., 2003) often make this analysis difficult. Ring widths are among the most used variables for climate reconstruction but they usually show a higher temporal instability in their relationship with climate than that of maximum latewood density (for the Pyrenees, see Büntgen et al., 2010).

The potential to analyze relatively long chronologies in the Mediterranean region has allowed for the reconstruction of the past climate (mainly precipitation and droughts). Several reconstructions of May-June precipitation have been performed, mainly over the last 300-400 yr, in a region including northern Greece, Turkey and Georgia: in northern Aegean-northern Anatolia a tree-ring network of oaks was used for reconstructing precipitation variability since 1089 CE (Griggs et al., 2007); in the Anatolian Peninsula a mixed conifer-broadleaf tree-ring network (mainly *P. nigra*, *P. sylvestris* and oaks; Akkemik et al., 2008), a *P. nigra* network (Köse et al., 2011) and a multi-species conifer network (mainly *P. nigra*, *P. sylvestris* and *Abies nordmanniana* (Steven) Spach; Köse et al., 2013) were used. In western Mediterranean, in central Spain, a higher frequency of exceptionally dry summers has been detected since the beginning of the 20th century using a mixed tree-ring network of *Pinus sylvestris* and *P. nigra* ssp. *salzmannii* covering the past four centuries (Ruiz-Labourdette et al., 2014), whereas a 800 yr temperature reconstruction from southeastern Spain using a site of *P. nigra* underlined predominantly higher summer temperatures during the transition between the Medieval Climate Anomaly (MCA) and the Little Ice Age (LIA) (Dorado Liñán et al., 2015). A recent reconstruction of spring-late summer temperature from the Pyrenees by means of a *P. uncinata* MXD network dating back to 1186 (Büntgen et al., 2017), underlines warm conditions around 1200 and 1400 and after 1850.

Reconstructions of past droughts and wet periods over the Mediterranean region have been created using climatic indices
such as the Standardized Precipitation Index (SPI; McKee et al., 1995) in Spain (modeling 12-month July SPI using several
species of the *Pinus* genre; Tejedor et al., 2016), and in Romania (modeling 3-month August standardized SPI using *P. nigra*;
Levanič et al., 2013), which allows for the identification of common large-scale synoptic patterns. Droughts have been
reconstructed using the Palmer Drought Severity Index (PDSI; Palmer 1965). Using actual and estimated multispecies tree-
ring data, Nicalut et al. (2008) found that the drought episodes at the end of the 20th century are similar to those in the 16th-
17th century for the western Mediterranean, whereas in the eastern parts of the region, the droughts seem to be the strongest
recorded in the past 500 yrs.

Early summer temperature has been reconstructed for 400 yr in Albania, from a *P. nigra* tree-ring network, finding stable
climate-growth relationships over time and a spatial extent of the reconstruction spanning over the Balkans and southern
Italy (Levanič et al., 2015). Currently, two summer temperature reconstructions close to the study area and based on
maximum latewood density (MXD) chronologies are available: (1) a reconstruction of AS temperature published by Trouet
(2014) covering the period 1675–1980 and centered on the northeastern Mediterranean-Balkan region includes sites from the
Italian Peninsula (used in this paper), the Balkan area, Greece and sites from the central and eastern European Alps to central
Romania and Bulgaria, the latter areas being characterized by continental climates; (2) a reconstruction of JAS temperature
published by Klesse et al. (2015), covering the period 1521–2010 and based on a chronology from Mt. Olympus (Greece).
As separate climate (temperature) reconstructions for northeastern Mediterranean-Balkan region including also Italy have
been published to date, the goal of this study was to collect dendrochronological data from Italian research groups and screen
the ITRDB for suitable data for climate reconstructions. We therefore investigate RW and MXD climate signals across Italy.
After carefully testing the climatic signals recorded in the tree-ring RW and MXD from different sites and different species,
the reconstruction that is proposed in this study is the first one including only forest sites from the Italian Peninsula.
Overall, the main objectives of this paper are:
(i) to identify the most important climatic drivers modulating tree-ring width (RW) and tree-ring maximum latewood density
(MXD) variability in forest sites from central and southern Italy. To our knowledge, this is the first attempt performed in
Italy with the clear objective to find common response patterns in conifer and broadleaf species using a multispecies tree-
ring network and site-specific historical climatic records;
(ii) to estimate the temporal stability of the climate-growth and climate-density relationships;
(iii) to perform a climatic reconstruction based only on trees *highly sensitive to climate* (HSTC); and
(iv) to estimate the spatial coherence of the obtained reconstruction in the region.

**2. Data and Methods**

**2.1 Study area and study sites**

The study region includes the whole Italian Peninsula and eastern Sicily and covers a latitudinal range from 37° 46' N to 44°
43' N (Fig. 1). The peninsula is roughly oriented NW-SE and its longitudinal axis is characterized by the Apennines that
reach their maximum altitude at their center (Corno Grande Mt., 2912 m a.s.l., Gran Sasso Massif); a higher altitude is
reached in eastern Sicily by the Etna Volcano (3350 m a.s.l.). The study region is surrounded by the Tyrrhenian and Adriatic
Seas and is characterized by a typical Mediterranean climate, with high temperatures and low precipitation during the
summer (from June to September), and by a Mediterranean-temperate regime at the higher altitudes of the Apennines (Fig.
2). Considering the climatic means at all the study sites (at a mean elevation of 1225±520 m a.s.l.) over the period 1880-
2014, the temperatures over the study region range from 0.2 °C (January) to 17.6 °C (in July and in August) and only 11 %
of the total annual precipitation falls during the summer (from June to August: 155 mm), whereas 34 % falls during winter

(from December$_{-1}$ to February: 493 mm). Autumn is the second wettest season (31 % of total annual precipitation) and spring is the third wettest (24 % of total annual precipitation) (Fig. 2).

The total forest cover in Italy, excluding the regions of the European Alps, is approximately 5.8 M hectares (Corpo Forestale dello Stato, 2005) which is 28 % of the considered surface. Forests characterize the landscape of the inner portion of the Apennine range, at mid to high elevations, and an additional 1.4 M hectares are covered by woodlands and shrublands, which are the so called Mediterranean 'macchia' that border the forests at low elevations and in areas relatively close to the sea. Overall, broadleaf species are much more abundant in the study region than conifer species, accounting for approximately ¾ of the forest cover (Dafis, 1997).

The study sites are distributed along the whole latitudinal range of the Italian Peninsula and tree-ring proxies include both RW and MXD series collected within the NEXTDATA project, from Italian Universities, and from the ITRDB (www.ncdc.noaa.gov site consulted on September 2015; see Table 1 for full bibliographic references). The dataset is based on 27 forest sites composed of several species (conifers at 16 sites, and broadleaves at 11 sites), from which tree-ring series of conifers (RW and MXD) and of broadleaves (RW) were prepared (Fig. 1, Table 1).

**2.2 Climate variables**

The availability of long and reliable time series of meteorological variables, possibly from stations located very close to forest sites, is crucial for estimating the climate-growth relationships. However, global or regional climatological datasets frequently lack local resolution, especially in remote sites. We, therefore, reconstructed synthetic records of monthly temperature and precipitation series to be representative of the sampled sites using the anomaly method (New et al., 2000; Mitchell and Jones, 2005), as described in Brunetti et al. (2012). Specifically, we reconstructed independently climatological normals (following the procedure described in Brunetti et al. (2014) and Crespi et al. (2017), by estimating a local temperature (precipitation) - elevation relationship, and exploiting a very high density data set from time series that are at least 30-year long). We also estimated the deviations from the normals by means of a weighted average of neighboring series, by exploiting the great amount of very long and high quality temperature and precipitation series available for Italy over the past 200/250 years (obtained from an improved version of Brunetti et al. (2006)). Finally, by the superposition of the two fields, we obtained temporal series in absolute values for each sampling site. The climate series start in different years due to data availability; however, most of the series start around the mid-19th century. Finally, in order to characterize meteorological drought conditions, we calculated the monthly Standardized Precipitation Index (SPI) at timescales of 1, 2, 3, 6, 9 and 12 months for all sites, based on the monthly values of precipitation, using the SPI_SL_6 code of the National Drought Mitigation Center at the University of Nebraska (http://drought.unl.edu).

**2.3 Chronology construction, climate sensitivity and climate reconstructions**

— *Raw data.* We examined all individual series of RW and MXD for correct dating using visual and statistical crossdating. In particular, we used statistical techniques to remove potential dating errors by comparing each individual series from one site against the mean site chronology, which was constructed excluding the analyzed individual series. Using the COFECHA software (www.ldeo.columbia.edu), the individual series were moved forward and backward 10 yr from their initial positions, and similarity indices were calculated over a 50-yr time window, thus highlighting the potential dating errors.

— *Site chronologies.* We used the Regional Curve Standardization approach (RCS; Briffa et al., 1992; Briffa and Melvin, 2011; Esper et al., 2003) both with the RW and MXD series to preserve the low-frequency variability in the site chronologies. We used the ARSTAN software (ver. 44 h3, www.ldeo.columbia.edu) and did not consider the pith offset estimates between the first measured ring and the actual first year of growth (Esper et al., 2009; Leonelli et al., 2016). The regional curve (RC) for the mean chronology, which was obtained after the series alignment to the first measured ring, was smoothed using a cubic spline with a width of 10 % of the chronology length (Büntgen et al., 2006). We computed ratios of

raw measurements vs. the values of growth predicted by the RC for all years of the individual series, and the resulting indexed series were averaged by a biweight robust mean to obtain the site chronologies of RW and of MXD. We constructed the RW and MXD site chronologies only for sites with at least 10 individual series fulfilling the following conditions: i) the individual series length was >100 yr; ii) the individual series correlation with the respective site chronology had r > 0.3; iii) the mean interseries correlation (MIC) had r > 0.3; and iv) the expressed population signal (EPS; Wigley et al., 1984; Briffa and Jones, 1990) was > 0.7. We used only the individual series fulfilling these conditions to construct the site chronologies. However, we accepted some exceptions in order to maximize the number of sites and chronologies available for analysis (see exceptions in Table 1).

— *Climate sensitivity*. We assessed species-specific climate sensitivity for the constructed RW and MXD site chronologies over the common period of 1880-1980 using correlation analysis and the site-specific monthly variables of temperature, precipitation and Standardized Precipitation Index, from March of the year prior to growth to September of the year of growth. We computed correlations using the DENDROCLIM software (Biondi and Waikul, 2004), applying a bootstrap with 1,000 iterations, and the obtained results were analyzed by grouping together conifer and broadleaf species.

— *Testing for climate-growth relationships at the site level*

To assess the influence of environmental settings on climate-growth relationships, for the conifer MXD site chronologies (i.e. the chronologies holding the strongest climatic signal; see Results), we performed a redundancy analysis (RDA) selecting as response variables the bootstrapped correlation coefficients of climate-growth relationships (Fig. 3) and as explanatory variables the environmental variables (geographical characteristics and climatic averages over the period 1880-1980). In order to attenuate co-variation within the environmental variables, we ran a PCA before the RDA and the following variables were finally chosen: Elevation (co-varying with Longitude: our sites are placed at higher elevation at increasing longitude (Table 1); average AS temperature; average JJA precipitation (co-varying with Latitude: higher latitude means higher precipitation amounts); average JJAS SPI_3 (at timescale of 3 months, i.e., the timescale resulting most significant; see Results). Moreover, for each of the MXD site chronologies, we calculated the Site Fitness (SF; Leonelli et al., 2016), representative of the percentage of selected HSTC series of conifer MXD with respect to the total of series available at each site.

We used the results of the climate sensitivity analysis to detect the *driving climate variables* (DCV; of temperature, precipitation and SPI) for each of the three groups of chronologies: MXD conifer, RW conifer and RW broadleaf. Specifically, for each group of chronologies and for each climate variable, we first identified the months with significant correlations at most sites (>50 %) and with mean correlation values of $|\overline{r}| > 0.25$ (black-filled squares in Fig. 3). Starting from the monthly climatic averages of the sites presenting significant correlations with these selected months, we constructed regional climate series by z-scoring the monthly series of each site and calculating regional mean departures; the series of each site were then completed over the maximum period covered by data and ri-converted in original units (based on regional mean departures and their specific means and standard deviations), and finally averaged between sites. We calculated the DCVs as means of two to four consecutive months of the regional series, except for August$_{-1}$ temperature vs. conifer RW (according to what was obtained in the site-level analysis of Fig. 3).

—*HSTC chronologies*. Based on the available RW and MXD indexed individual series from all of the sites, we constructed six HSTC chronologies, as in Leonelli et al. (2016). However, given the smaller number of datasets available in this study and the shortness of the time series, a modified version of the method was applied. Specifically, we tested all of the RW (conifer and broadleaf) and MXD (only conifer) indexed individual series against each of the above-defined six DCV, and we used only the individual tree-ring indexed series with correlation values of $|\overline{r}| > 0.25$ in both of the 100 yr subperiods of the climatic dataset (1781-1880 and 1881-1980) for building each of the six HSTC chronologies (which was done by simply averaging together the selected indexed series). We constructed the six HSTC chronologies starting from all of the indexed

individual series of conifer MXD (148 series), of conifer RW (245) and of broadleaf RW (140), which were previously obtained, while constructing the site chronologies (also, the indexed individual series from sites not meeting the fixed quality standards for a site chronology were included at the beginning of the selection).

— *Climate sensitivity through time.* To test the stability of the climate signals recorded in the HSTC chronologies, we conducted a moving correlation analysis between the six HSTC chronologies and their respective DCV, computing bootstrapped correlation coefficients with DENDROCLIM over 60 yr time windows that were moved one year per iteration over the longest available periods.

— *Climate reconstruction.* We used only the HSTC chronology showing the highest absolute values of correlation and the most stable signal over time (i.e., the conifer MXD for late summer temperature; see Results) for the climate reconstruction. To extend this HSTC chronology as far back in time as possible, we also added the oldest available individual MXD indexed series with correlations of $|\overline{r}| > 0.25$ with this chronology and that had a minimum length of 100 yr. For constructing the chronology for climate reconstruction we applied an arithmetic mean to the indexed series, after having normalized all individual series over the common period 1879-1962. Moreover, to account for the changing sample size through time, a variance stabilization of the resulting chronology was performed using Briffa's RBAR- weighted method (Osborn et al., 1997). In order to improve the HSTC chronology over the early period showing an EPS < ~0.8 (i.e. before 1713 in the first version of the HSTC chronology), we considered the yearly difference of the indexed normalized series from the mean and discarded the early portion of the series exceeding the threshold of 2.5 standard deviations in a given year (one series was truncated at 1713, whereas the other nine felled within a common variability). Finally, we re-normalized all series and recalculated the final version of the HSTC chronology used for the temperature reconstruction as described above. We calibrated and verified linear regression and scaling models (Esper et al., 2005) over the 100 yr periods 1781-1880 and 1881-1980, respectively, and then the same was done over the inverted periods, in order to estimate model performances and stability. We computed Reduction of Error (RE; Fritts, 1976) and Coefficient of Efficiency (CE; Briffa et al., 1988) statistics to assess the quality of the reconstructions. We then used the reconstructed series of late summer temperatures over the period 1901-1980 to build a spatial correlation map with the KNMI Climate Explorer (https://climexp.knmi.nl/; Trouet and Oldenborgh, 2013), using the 0.5° grid of August-September average temperature and of AS average precipitation (CRU TS 4.0, Climatic Research Unit, University of East Anglia Harris et al., 2014). We used this independent dataset instead of the Italian one, as our primary goal was to analyze how far from the Italian Peninsula the reconstructed climatology is still representative.

## 3 Results

— *Site chronologies.* We obtained fifteen RW site chronologies (11 from conifers and 4 from broadleaves) and eight MXD site chronologies (from conifers) and we used them to estimate climate sensitivity at the site level and to detect the most important climatic drivers over the study region (for species percentages, see boxes in Fig. 3A, 3A' and 3A''). We performed the construction of the HSTC chronologies (for the analysis of the temporal stability of climate signals and for climate reconstruction) using also the individual series from the twelve sites (5 from conifer and 7 from broadleaves; see Table 1, grey-shaded areas in Table 2 and Methods) for which the site chronologies did not meet the quality standards. The maximum time span of tree-ring data covers the period from 1415 (ITRDBITAL015) to 2013 (QFIMP1 and QFIMP2). However, the mean chronology length is 215±130 yr for conifers and 175±25 yr for broadleaves (values rounded to the nearest 5 yr; Table 2). Over the common period considered (1880-1980 for all MXD and RW chronologies), the mean series intercorrelation and expressed population signal are approximately 0.5 and 0.8, respectively.

— *Tree-ring sensitivity to climate.* The site-specific sensitivity analysis performed over the common period of 1880-1980 revealed that MXD in conifers records stronger climatic signals than RW in either conifers or broadleaves, in terms of the

average correlation coefficient, the number of months showing statistically significant values ($p < 0.05$) and the fraction of chronologies (over the maximum number available) responding to the same climatic variable (Fig. 3). In particular, all conifer MXD chronologies were found to be positively influenced by late summer temperatures (August and September), whereas precipitation from June to August is negatively correlated with most of them (Fig. 3A and 3B). In terms of SPI, the highest correlations (for both MXD and RW) were obtained for the indices calculated at the timescales of 2 and mainly of 3 months (SPI_3; only the latter is reported in the Results), while longer timescales showed fewer significant correlation values. Most conifer MXD were found to be negatively correlated with SPI_3 from June to September, highlighting that low index values, i.e., drought periods, are associated with high MXD in the tree rings, and vice versa (Fig. 3C).

For conifer RW, significant correlation coefficients, i.e., those exceeding the mean value of $|\overline{r}| > 0.25$ for more than 50 % of the available chronologies, were obtained only for the August temperatures of the year prior to growth (a negative correlation; Fig. 3A'). In the other months, correlations are generally low and sometimes show opposite signs for the same climatic variable. However, a slightly stronger influence from the climatic variables for the summer months prior to growth can be noted (black areas in Fig. 3A', 3B' and 3C').

Broadleaf RW were found to be positively influenced by high precipitations and low drought occurrences (high SPI_3 values) during the summer months (June and July precipitation and June to August SPI_3; Fig. 3B" and 3C"), whereas the temperature did not show a significant influence (Fig. 3A").

— *Influence of environmental settings on climate-growth relationships & Site Fitness.* We found that the strength of the AS signal correlated positively with latitude (mean precipitation) and negatively with elevation (longitude) (Fig. 4A). Summer precipitation amounts and elevation correlated negatively in our dataset of MXD, revealing the dominance of the latitudinal gradient of larger precipitation in northern areas over the expected altitudinal gradient of higher precipitation at higher altitudes: sites in northern areas, even if at lower altitudes, receive more summer precipitation than sites in southern regions at higher altitude. The RDA analysis revealed that both parameters were on opposing sides of the first two axes explaining 89.55 % of the variance of the dataset: the F1 axis alone explains up to 72% of the variance in response variables, and especially in AS temperature and JJAS SPI_3 signals. Concerning Site Fitness, especially sites located at higher latitudes, in particular northern of 42° N (all of *Abies alba*) presented values of SF > 80%, and up to 86% (Fig. 4B). South of 42° N, all sites (including also two sites of *Abies alba*) presented a SF of approximately 10%, with the *Pinus leucodermis* site showing the highest SF value (52%) and a *P. nigra* site the lowest (0 %).

— *Stability of the climatic signal over time.* The six comparisons performed between the HSTC chronologies and the DCV were deemed important to understand the influence of temporal climatic variability on conifers MXD and RW and on broadleaf RW (Fig. 5). The moving-window correlation analysis revealed that the HSTC conifer MXD chronology held the strongest and most stable climatic signal of late summer temperature over time, with values of correlation coefficient ranging from approximately 0.4 to nearly 0.8 in the more recent periods analyzed (#1 in Fig. 5). In the other two HSTC chronologies based on conifer MXD (#2 and #3 in Fig. 5), starting from the time window 1881-1940 up to recent periods, we always found higher absolute values for SPI_3 than for precipitation, with values of correlation reaching approximately -0.7 and -0.6, respectively, (#3 and #2 in Fig. 5). For the conifer RW, a strong change in the temperature signal of August prior to growth was found (#4 in Fig. 5), with correlation values shifting from positive (and statistically non-significant) in the early period of analysis to negative (approximately -0.5) in the mid to late analysis period. The two HSTC chronologies of broadleaf RW showed nearly the same correlation values and similar patterns with both the June and July precipitation and the June to August SPI_3, with values at approximately +0.5 (#5 and #6 in Fig. 5).

— *Climate reconstruction.* The reconstruction of the late summer temperature for the Italian Peninsula was, therefore, based on the HSTC chronology of conifer MXD, while the conifer RW chronology was disregarded due to its low signal stability over time. The reconstructed series based on the scaling approach starts in 1657 and has a minimum sample replication of ten trees since 1713 (Fig. 6A); it reproduces well the variability of the instrumental record and underlines the periods of climatic

cooling (and likely also wetter conditions) in the years 1699, 1740, 1814, 1914, 1938. The low-pass filtered series emphasize the mid-length fluctuations and show evidence of periods of temperature underestimations (centered around 1799, 1925 and 1952) and of overestimations (around 1846) (Fig. 6B); however, the differences from the instrumental record were always found to be within 1° C for both scaling and regression approaches. The two models tended to have higher values when they were calibrated over the period 1781-1880 and lower values when they were calibrated over the period 1881-1980 (Table 3). The CE statistics showed similar patterns of RE and its values were always positive for both the regression and the scaling model.

— *Spatial coherence of the reconstruction.* The spatial coherence of the late summer temperature reconstruction of the Italian Peninsula performed over the Mediterranean region indicated that, for the period of 1901-1980 (defined by the beginning of the CRU TS 4.0 climate series and the end of the MXD series), the reconstructed temperature series matched very well the temperature variability in Italy south of the Po Plane, Sardinia and Sicily and the western Balkan area (r > 0.6). Correlations above 0.4 were still found throughout the Alpine arc, the central Balkan, western Anatolia, as well as in northwestern Maghreb. (Fig. 7A, 7B). In detail, the reconstructed temperature highly correlated westward up to Sicily and Sardinia, and eastward to the western Balkan area along the Adriatic Sea up to northern Greece, whereas r values were already lower than 0.5 in a wide arch including northern Tunisia, southern France, the inner range of the European Alps, Turkey and southern Anatolia. The reconstructed AS temperature series significantly correlated also with mean AS precipitation, especially in a wide belt between 35° and 50° N of latitude centered over Croatia (negative correlations, below -0.6) and the Balkan region up to the Black Sea. For Italy, correlations above 0.4 were found in the southern portion of the Peninsula, whereas weaker correlations were found westward up to the eastern Pyrenees and northern Maghreb. Positive correlations, above 0.3, were found in a belt in northern Europe at approximately 55° N of latitude, centered over Ireland, Scotland and Wales, and up to Denmark and the southern Scandinavian Peninsula (Fig. 7C, 7D).

## 4 Discussion

The climate signals recorded in the multispecies and multiproxy tree-ring network from the Italian Peninsula revealed a general coherence with other climate-growth analyses performed in Mediterranean environments. As found in the Pyrenees for a conifer tree-ring network (Büntgen et al., 2010), we found generally strong and coherent signals in MXD, independent of species. In particular, in our record, the late summer temperature was well recorded in MXD chronologies, and the correlations with climate were stable over time. The MXD chronologies were mainly related to temperature; however, we found clear signals of the influence of summer precipitation and droughts. In the Mediterranean area, especially during summer, high temperature is often associated with low precipitation and drought; therefore, when interpreting the temperature reconstructions based on tree-ring MXD in the Mediterranean area, also the associated influence of precipitation and droughts on MXD should be taken in account. The SPI, which was used here to represent drought conditions, was found to have higher correlations with both MXD and RW for the index calculated at the timescales of 2 and mainly of 3 months, whereas lower correlations were found at lower (1 month) and higher (6, 9 and 12 months) timescales. Thus, trees respond to the drought signal at this time scale, which reflects soil moisture droughts in the root zone (the SPI_3 is also the index used for modeling agricultural droughts, see e.g., WMO, 2012). On the contrary, trees apparently do not respond strongly to the signal of hydrological droughts at the catchment level (SPI at timescales of above 6 months).

The reconstructed series of the late summer temperatures for the Italian Peninsula were shown to have a strong coherence with the instrumental record and with both the reconstruction of AS temperature proposed by Trouet (2014) for the northeastern Mediterranean-Balkan region, and of JAS temperature proposed by Klesse et al. (2015), (Fig. 6C and Table 4). The three reconstructions are highly consistent, and the reconstruction of Trouet (2014) also includes the sites used in this paper. However, there are some differences between the Trouet's (2014) reconstruction and the one presented here: our reconstructed AS temperature in the Italian Peninsula tends to generally show less negative fluctuations over time than the

reconstruction from the Balkan area. While common periods of climatic cooling were recorded in both areas in 1741 and 1814, similar events were seen in 1913 and in 1977 only in the Balkan area. Interestingly, the periods of the larger differences between the reconstructed AS temperature and the instrumental record (around 1799, 1846, 1925 and 1952) are also those with strong coherence between the two reconstructions, suggesting a regional consistency in the responses to climate, possibly facilitated by similar precipitation patterns in the two regions during late summer. We also compared all these tree-ring based temperature reconstructions (of AS and JAS) with the summer (JJA) temperature gridded dataset of Luterbacher et al. (2004) (based on proxy, documentary, and instrumental data), for the gridpoints containing our MXD sites and over the common period covered by instrumental data from Italy used in the present work, i.e. since 1763 (Online Material 1). Both the instrumental data for Italy and the proxy-based reconstructions showed a good coherence with Luterbacher et al. (2004) at the decadal scale, however in the 1790-1810 period they showed opposite trends (with generally lower temperatures than in Luterbacher et al. (2004)) and more marked negative fluctuations in the 1810s.

Contrary to what was found in our reconstruction and in the northeastern Mediterranean, another late summer temperature reconstruction from Corsica, based on tree-ring stable carbon isotopes (Szymczak et al., 2012), shows periods of high temperature at the end of 1600 and beginning of 1700 and a very slight cooling during the 1810s, probably owing to the effect of the surrounding seas.

An important factor influencing the tree-ring MXD is volcanism, especially in correspondence of highly explosive eruptions that can change the intensity of the incoming solar radiation and that are able to change circulation patterns and cool the climate at hemispheric to global scale (e.g., Briffa et al., 1998). The largest explosive eruptions (Volcanic Explosivity Index $\geq 6$; Siebert et al., 2011) correspond to local minimum densities in the tree rings (Fig. 6C and 6D), and some of them are well known to be associated with years of famine and low crop yields. The year 1699 and the proceeding decades are known for being years of recurrent explosive eruptions in Iceland and Indonesia (Le Roy Ladurie, 2004), inducing great famines around Europe and North America (Mitchison, 2002). The 1809 eruption of unknown source (Guevara-Murua et al., 2014) and the 1815 eruption of Mount Tambora induced a decade of very low summer temperature and high precipitation (Luterbacher and Pfister, 2015). This was the coldest decade of the so called Little Ice Age (Lamb, 1995), corresponding also to glacier advance phases in the Alps, that reached their first maximum extent of the Holocene (the second and last, was around 1850; e.g., Matthews and Briffa, 2005). Eruptions of Mount Krakatoa in 1883 and of Novarupta (Aleutian Range) in 1912 correspond to local minima in the MXD. But a straightforward correspondence between minimum values of MXD densities and large eruption is lacking: some differences at regional scale with respect to global scale may occur owing to local circulation patterns and/or the presence of seas, as it is the case of the 1783 Grímsvötn Volcano eruption (Iceland), that corresponds to unexpected high MXD densities in tree rings from the Mediterranean area (Fig. 6) but not at the global scale (see Fig. 1 in Briffa et al., 1998), or the local minimums of MXD density of 1740 and 1938 found in this paper that are not linked to any particular large eruption.

The Apennines and the European Alps often show similar annual changes in precipitation amounts. However, in some periods, they show opposite decadal trends, such as after 1830, when precipitation was increasing in northern Italy but decreasing in the South, and after 2000, when the opposite behavior was observed (Brunetti et al., 2006). In the Italian Peninsula, the summer (JJA) and the autumn (SON) precipitation in 1835-1845 showed local minimum values in the instrumental record, likely inducing higher densities in the tree-ring latewood and, therefore, overestimations in model temperature values (Fig. 6B). Moreover, uncertainties between the instrumental records and MXD may rise given that trees do not respond linearly to high temperatures, resulting in a divergence between climatological and MXD records (e.g., for the Alps and Europe, Battipaglia et al., 2010). As found in this study, MXD is influenced by both late-summer temperature and summer precipitation and drought. In the Mediterranean, these variables are usually negatively correlated. Therefore, in some periods, a given value of MXD could have been caused either by temperature and less by drought or vice versa. Of the considered explanatory environmental variables, it is especially the latitudinal regime of summer precipitation that

modulates the MXD sensitivity to AS temperature and to summer drought (Fig. 4A): sites in northern Italy (more mesic and at lower elevation) show stronger climate signals than sites in the southern areas (more xeric and at higher elevation). In addition to the stronger AS temperature influence on MXD in the northern chronologies, the effect of summer precipitation/drought becomes equally stronger at the southern sites. MXD sites from southern Italy present a markedly lower SF than sites from central-northern Apennines. Considering the responses related to the type of species that in our dataset the influence of AS temperature on MXD in *A. alba* is more affected by summer precipitation amounts than in *P. lucodermis* and *P. nigra*. On the other hand, the influence of summer drought on MXD in pines is more affected by elevation.

Climatic signals recorded in RW tree-ring chronologies of conifers and broadleaves showed fewer clear common patterns in their correlations with climate variables than conifer MXD, although some climatic signals, which were valuable for climate reconstructions and for understanding climate impacts on tree-ring growth, were detected. In our records, the summer drought signal was clearly recorded at all broadleaf sites (Fig. 3C''), with moist periods (low recurrence of drought, i.e., high SPI_3 values) positively affecting tree-ring growth. The drought signal (as well as the precipitation signal) was fairly stable over time (#6 and #5 in Fig. 5), suggesting the possibility for climate drought (and precipitation) reconstructions in the Italian Peninsula with the availability of longer dendrochronological series. Differently from Levanič et al. (2015), we did not find a stable signal in conifer RW associated with the temperature signal, even though our correlations are related only to $August_{-1}$ temperatures (#4 in Fig. 5). The signal of previous August temperatures recorded in conifer chronologies (Fig. 3A') is too much variable through time to allow for a reconstruction (Fig. 5). Here, the change in sensitivity is probably related to the negative effect of droughts in summer and autumn (June to October) prior to growth (see SPI_3 correlations; Fig. 3C'). The question of the temporal stability of climate-growth relationships is sometimes underestimated in climate reconstructions, even though changes of climate signals over time have been identified in the Mediterranean region (Lebourgeois et al., 2012; Castagneri et al., 2014) and in the European Alps (Leonelli et al., 2009; Coppola et al., 2012).

Tree-ring growth may be affected also by large-scale climate variability, such as the North Atlantic Oscillation (NAO), the prominent mode of atmospheric circulation in the North Atlantic that affects temperature and precipitation patterns in Europe (D'Arrigo et al., 1993; Cook et al., 2002). In the eastern Mediterranean region, a teleconnection with summer climate conditions in the British Isles has been found in a summer temperature reconstruction for Bulgaria (Trouet et al., 2012), where tree-ring growth patterns are strongly linked to drought conditions. For Greece and the region eastward (Klesse et al., 2015), a prominent dipole pattern of summer NAO was found, whereas in Italy a major effect on tree growth was found for winter NAO, that correlates negatively with winter precipitation, that in turn determines soil moisture during the growing season (Piovesan and Schirone, 2000). Temporal instabilities of tree growth with climatic variables may be linked to several environmental and physiological factors that may influence tree growth processes and tree-ring sensitivities to climate, such as the still-debated fertilization effect due to increasing $CO_2$ concentration in the atmosphere (e.g., Brienen et al., 2012). On the other hand, biomass production and tree growth in Mediterranean forests seem to be linked to nutrient availability and environmental constraints rather than to the availability of $CO_2$ (e.g., Jacoby and D'Arrigo, 1997; Körner, 2003; Palacio et al., 2013). Local low-energy geomorphological processes such as sheetfloods (e.g., Pelfini et al., 2006) may impact tree-ring growth as well as the presence of an active volcano and its direct influence on local climate and atmospheric conditions (such as the Vesuvio Volcano, Battipaglia et al., 2007, or the Etna Volcano, Sailer et al., 2017), or air/soil pollution linked to $SO_2$, $NO_2$, or $O_3$ depositions and dust depositions from industrial plants or mines (in central Europe; Elling et al., 2009, Kern et al., 2009; Sensula et al., 2015): all these environmental factors may lower the tree-ring sensitivity to climate. Emissions from car traffic may also alter the tree-ring stable isotope signals and the related climatic signals (Saurer et al., 2004; Leonelli et al., 2012). The species-specific physiological responses of tree growth to climate variability may be nonlinear when high summer temperatures and low soil moistures exceed specific physiological thresholds, and can interrupt tree-ring growth during the growing season in Mediterranean climates (Cherubini et al., 2003). In terms of ecological factors, the

recurrent attacks of defoliator insects (e.g., the pine processionary moth; Hódar et al., 2003), the occurrence of forest fires (e.g., San-Miguel-Ayanz et al., 2013) or herbivory grazing and land abandonment (Herrero et al., 2011; Camarero and Gutiérrez, 2004) may influence vegetation dynamics and tree growth in Mediterranean forests, thus potentially introducing non-climatic effects into the chronologies.

Our reconstruction of the late summer temperature based on conifer MXD shows a clear stable climatic signal over time, and we could define the spatial coherence of the temperature reconstruction, thus allowing for the determination of the regions that could be included to extend the reconstruction further back in time. The late-summer temperature reconstruction of Trouet (2014) is more appropriate for the region around the southern and inner Balkans; our reconstruction is the first fully coherent late summer temperature reconstruction for Mediterranean Italy, extending in a west-east direction from Sardinia and Sicily to the Western Balkan area. As also evidenced by the site-level analysis, MXD depends also on precipitation and drought (Fig. 3B and 3C), especially in southern sites: our late-summer temperature reconstruction negatively correlates also with late summer precipitation, more in southern than in the central and northern Italian Peninsula, in the whole Balkan region up to the Black Sea, and especially in a region centered over Croatia. By contrast, it is positively correlate with precipitation in Ireland and Scotland and southern Scandinavia. This spatial approach allows for the definition of areas responding to climatic forcing in homogenous ways, which may also help predict the forest response to future climate change in the Mediterranean region.

**5 Conclusion**

The climate sensitivity analysis of a multispecies RW and MXD tree-ring network from the Italian Peninsula reveals that conifer MXD chronologies record a strong and stable signal of late summer temperatures and, to a lesser extent, of summer precipitation and drought. In contrast, the signals recorded by both conifer and broadleaf RW chronologies are less stable over time but are still linked to the summer climates of the year prior to growth (conifer) and the year of growth (broadleaves). The MXD sensitivity to AS temperature and to summer drought is mainly driven by the latitudinal gradient of summer precipitation amounts, with sites in northern areas (above 42° N, all silver fir sites, at lower altitudes) showing stronger climate signals than sites in the South (below 42°N, mainly *P. leucodermis* and silver fir sites at higher altitudes).

The reconstruction of the late summer temperatures over the past 300 yr (up to 1980), based on the conifer MXD chronologies, shows a strong coherence with the reconstruction performed by Trouet (2014) for the northeastern Mediterranean-Balkan region and by Klesse et al. (2015) for Greece and the eastward region. With respect to the former reconstruction, however, the temperatures reconstructed in our study show less negative fluctuations during the last century, likely because all of our sites are located along the Italian Peninsula and are relatively close to the sea. According to our reconstruction, 1699, 1740, 1814, 1914, and 1938 were years of particularly low late summer temperatures over the study region (with some of them linked to large volcanic eruptions affecting climate at the global scale), whereas the highest temperature was found in 1945. The late-summer temperature reconstruction proposed here is representative of a wide area covering the Italian Peninsula, Sardinia, Sicily and the Balkan area close to the Adriatic Sea. These areas could be considered to further enhance the regional reconstruction discussed here. Moreover, this reconstruction is correlated also with late-summer precipitation in the central Mediterranean and the Balkan region, thus further helping in better assessing climate change impacts on forests in homogenous areas within the Mediterranean climate change hot spot.

**Data availability**. Data will be available in the Online Material 2.

**Competing interests.** The authors declare that they have no conflict of interest.

**Acknowledgements.** This study was funded by the project of strategic interest NEXTDATA (PNR National Research Programme 2011-2013; project coordinator A. Provenzale CNR-IGG, WP leader V. Maggi UNIMIB and CNR-IGG), and by the following PRIN 2010-2011 projects (MIUR - Italian Ministry of Education, Universities and Research): grant no. 2010AYKTAB_006 (national leader C. Baroni), and grant no. B21J12000560001 'CARBOTREES'.

This study is also linked to activities conducted within the following COST Actions (European Cooperation in Science and Technology), financially supported by the EU Framework Programme for Research and Innovation HORIZON 2020: FP1106 'STReESS' (Studying Tree Responses to extreme Events: a SynthesiS), and CA15226 CLIMO (Climate-Smart Forestry in Mountain Regions). We thank the several researchers who uploaded their raw data onto the ITRDB, the two anonymous reviewers and Prof. Luterbacher for their useful comments and suggestions.

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

**Table 1:** References for all the dendrochronological data used in this research, information on site locations, types of parameter used at each site and the tree species. Sites are ordered along a decreasing latitudinal gradient, after differentiating between conifers and broadleaves (dotted line).

| | | Database information and site location | | | | | | Type of tree-ring parameter | | | |
|---|---|---|---|---|---|---|---|---|---|---|---|
| Dataset name | Database Source | Original contributor | Bibliographic reference | Location Name | Latitude N | Longitude E | Elevation (m a.s.l.) | RW chr. | RW series | MXD chr. | Species |
| ITRDBITAL017 | ITRDB | Ori, G.G. | *https://www.ncdc.noaa.gov/paleo/study/4079* | Monte Cantiere | 44° 16' 48'' | 10° 48' 00'' | 800 | x | | | *Pinus sp.* |
| ITRDBITAL009 | ITRDB | Schweingruber, F.H. | *https://www.ncdc.noaa.gov/paleo/study/4301* | Abetone | 44° 07' 12'' | 10° 42' 00'' | 1400 | | x | x | *Abies alba* |
| ITRDBITAL004 | ITRDB | Biondi, F. | *https://www.ncdc.noaa.gov/paleo/study/2753* | Campolino | 44° 06' 45'' | 10° 39' 44'' | 1650 | | x | | *Picea abies* |
| ITRDBITAL008 | ITRDB | Schweingruber, F.H. | *https://www.ncdc.noaa.gov/paleo/study/4540* | Mount Falterona | 43° 52' 12'' | 11° 40' 12'' | 1450 | x | | x | *Abies alba* |
| ITRDBITAL003 | ITRDB | Biondi, F. | *https://www.ncdc.noaa.gov/paleo/study/2760* | Pineta San Rossore | 43° 43' 12'' | 10° 18' 00'' | 5 | x | | | *Pinus pinea* |
| ITRDBITAL022 | ITRDB | Becker, B. | *https://www.ncdc.noaa.gov/paleo/study/2706* | Pratomagno Bibbiena - Appennini | 43° 40' 12'' | 11° 46' 12'' | 1050 | | x | | *Abies sp.* |
| ITRDBITAL012 | ITRDB | Schweingruber, F.H. | *https://www.ncdc.noaa.gov/paleo/study/4374* | Ceppo Bosque di Martense | 42° 40' 48'' | 13° 25' 48'' | 1700 | x | | x | *Abies alba* |
| Abies-Abeti-Soprani | UNIMOL | | | Colle Canalicchio-Abeti Soprani | 41° 51' 40'' | 14° 17' 51'' | 1350 | | x | | *Abies alba* |
| ITRDBITAL016 | ITRDB | Schweingruber, F.H. | *https://www.ncdc.noaa.gov/paleo/study/4536* | Monte Mattone | 41° 46' 48'' | 14° 01' 48'' | 1550 | x | | x | *Pinus nigra* |
| ITRDBITAL001 | ITRDB | Biondi, F. | *https://www.ncdc.noaa.gov/paleo/study/2752* | Camosciara Mt. Amaro | 41° 46' 12'' | 13° 49' 12'' | 1550 | x | | | *Pinus nigra* |
| ITRDBITAL002 | ITRDB | Biondi, F. | *https://www.ncdc.noaa.gov/paleo/study/2759* | Parco del Circeo | 41° 19' 48'' | 13° 03' 02'' | 5 | x | | | *Pinus pinea* |
| AAIBA | UNIBAS | | | Ruoti (PZ) | 40° 42' 04'' | 15° 43' 43'' | 925 | | x | | *Abies alba* |
| ITRDBITAL011 | ITRDB | Schweingruber, F.H. | *https://www.ncdc.noaa.gov/paleo/study/4541* | Mount Pollino | 39° 54' 00'' | 16° 12' 00'' | 1720 | x | | x | *Abies alba* |
| ITRDBITAL015 | ITRDB | Schweingruber, F.H. | *https://www.ncdc.noaa.gov/paleo/study/4644* | Sierra de Crispo | 39° 54' 00'' | 16° 13' 48'' | 2000 | x | | x | *Pinus leucodermis* |
| ITRDBITAL010 | ITRDB | Schweingruber, F.H. | *https://www.ncdc.noaa.gov/paleo/study/4420* | Gambarie Aspromonte | 38° 10' 12'' | 15° 55' 12'' | 1850 | x | | x | *Abies alba* |
| ITRDBITAL013 | ITRDB | Schweingruber, F.H. | *https://www.ncdc.noaa.gov/paleo/study/4304* | Aetna Linguaglossa | 37° 46' 48'' | 15° 03' 00'' | 1800 | x | | x | *Pinus nigra* |
| ITRDBITAL019 | ITRDB | Nola, P. | *https://www.ncdc.noaa.gov/paleo/study/4042* | Corte Brugnatella | 44° 43' 12'' | 09° 19' 12'' | 900 | x | | | *Quercus robur* |
| Fagus-Parco-Abruzzo | UNIMOL | | | Val Cervara | 41° 49' 00'' | 13° 43' 00'' | 1780 | | x | | *Fagus sylvatica* |
| Fagus-Gargano | UNIMOL | | | Parco Nazionale del Gargano Riserva Pavari | 41° 49' 00'' | 16° 00' 00'' | 775 | | x | | *Fagus sylvatica* |
| Fagus-Montedimezzo | UNIMOL | | | Riseva MaB Unesco Collemeluccio-Montedimezzo | 41° 45' 00'' | 14° 12' 00'' | 1100 | x | | | *Fagus sylvatica* |
| Cervialto-FASY | UNINA2 | | | Monti Picentini | 40° 50' 23'' | 15° 10' 03'' | 800 | | x | | *Fagus sylvatica* |
| Fagus-Cilento | UNIMOL | | | Parco Nazionale del Cilento Ottati | 40° 28' 00'' | 15° 24' 00'' | 1130 | | x | | *Fagus sylvatica* |
| QCIBG | UNIBAS | | | Gorgoglione (MT) | 40° 23' 09'' | 16° 10' 04'' | 820 | | x | | *Quercus cerris* |
| QFIMP1 | UNIBAS | | | San Paolo Albanese (PZ) | 40° 01' 20'' | 16° 20' 26'' | 1050 | x | | | *Quercus frainetto* |
| QFIMP2 | UNIBAS | | | Oriolo (CS) | 40° 00' 10'' | 16° 23' 29'' | 960 | x | | | *Quercus frainetto* |
| Fagus-Sila | UNIMOL | | | Parco Sila | 39° 08' 00'' | 16° 40' 00'' | 1680 | | x | | *Fagus sylvatica* |
| Fagus-Parco-Aspromonte | UNIMOL | | | Aspromonte | 38° 11' 00'' | 15° 52' 00'' | 1560 | | x | | *Fagus sylvatica* |
| | | | | | | | 1235 mean elevation | 15 sites | 12 sites | 8 sites | |

**Table 2:** Main characteristics of the chronologies used in this research, separating RW (comprised of both broadleaf and conifer
species) and MXD (only conifer species). For each site and parameter, the total number of series available and the number of
series showing a correlation value $0.2 < r < 0.3$ with the respective master chronology is reported. Grey-shaded areas depict
values that do not exceed the fixed thresholds of MIC > 0.3, EPS > 0.7 and a number of series > 10, determining the exclusion of
the chronology from further analyses. Sites ordered as in Table 1.
a = Mean Interseries Correlation of raw series, calculated using the maximum period available at each site.
b = Expressed Population signal of indexed series in the common period of 1880-1980.
* series up to 80 yr included.
** chronology built with less than 10 series (good EPS).
*** common period with later Start date or earlier End date.
**** sites without chronology [....] are not included in the computation.

| Dataset name | RW series characteristics | | | | | | | MXD series characteristics on the maximum period available | | | | | | |
|---|---|---|---|---|---|---|---|---|---|---|---|---|---|---|
| | Start date | End date | Time span | MIC[a] | EPS[b] | # series | # series 0.2< r <0.3 vs. master | Start date | End date | Time span | MIC1 | EPS2 | # series | # series 0.2< r <0.3 vs. master |
| ITRDBITAL017 | 1856 | 1989 | 134 | 0.43 | 0.76 | 14 | 0 | - | - | - | - | - | - | - |
| ITRDBITAL009 | [1846] | [1980] | [135] | [0.73] | [0.66] | 13 | 0 | 1846 | 1980 | 135 | 0.76 | 0.86 | 21 | 0 |
| ITRDBITAL004 | [1836] | [1988] | [153] | [0.51] | [0.49] | 11 | 0 | - | - | - | - | - | - | - |
| ITRDBITAL008 | 1827 | 1980 | 154 | 0.62 | 0.70 | 12 | 0 | 1827 | 1980 | 154 | 0.66 | 0.87 | 12 | 0 |
| ITRDBITAL003*; ** | 1861 | 1988 | 128 | 0.51 | 0.72 | 9 | 0 | - | - | - | - | - | - | - |
| ITRDBITAL022*** | [1539] | [1972] | [434] | [0.45] | [0.67] | 6 | 1 | - | - | - | - | - | - | - |
| ITRDBITAL012 | 1654 | 1980 | 327 | 0.57 | 0.85 | 26 | 0 | 1654 | 1980 | 327 | 0.59 | 0.91 | 25 | 0 |
| Abies-Abeti-Soprani* | [1838] | [2005] | [168] | [0.53] | [0.50] | 11 | 0 | - | - | - | - | - | - | - |
| ITRDBITAL016 | 1844 | 1980 | 137 | 0.54 | 0.84 | 17 | 0 | 1844 | 1980 | 137 | 0.43 | 0.75 | 15 | 0 |
| ITRDBITAL001 | 1750 | 1987 | 238 | 0.52 | 0.77 | 16 | 0 | - | - | - | - | - | - | - |
| ITRDBITAL002* | 1878 | 1988 | 111 | 0.51 | 0.72 | 16 | 0 | - | - | - | - | - | - | - |
| AAIBA* | [1866] | [2007] | [142] | [0.51] | [0.55] | 13 | 0 | - | - | - | - | - | - | - |
| ITRDBITAL011 | 1800 | 1980 | 181 | 0.58 | 0.85 | 20 | 0 | 1800 | 1980 | 181 | 0.54 | 0.84 | 18 | 0 |
| ITRDBITAL015 | 1415 | 1980 | 566 | 0.58 | 0.95 | 22 | 0 | 1441 | 1980 | 540 | 0.50 | 0.76 | 21 | 0 |
| ITRDBITAL010 | 1790 | 1980 | 191 | 0.53 | 0.76 | 19 | 0 | 1790 | 1980 | 191 | 0.50 | 0.85 | 18 | 0 |
| ITRDBITAL013 | 1773 | 1980 | 208 | 0.57 | 0.88 | 20 | 0 | 1795 | 1980 | 186 | 0.44 | 0.78 | 18 | 0 |
| ITRDBITAL019 | 1779 | 1989 | 211 | 0.54 | 0.82 | 16 | 0 | - | - | - | - | - | - | - |
| Fagus-Parco-Abruzzo | [1716] | [2008] | [293] | [0.36] | [0.73] | 3 | 0 | - | - | - | - | - | - | - |
| Fagus-Gargano | [1821] | [2009] | [189] | [0.23] | [0.42] | 3 | 3 | - | - | - | - | - | - | - |
| Fagus-Montedimezzo | 1844 | 2005 | 162 | 0.67 | 0.85 | 15 | 0 | - | - | - | - | - | - | - |
| Cervialto-FASY | [1828] | [2003] | [176] | [0.39] | [0.52] | 10 | 0 | - | - | - | - | - | - | - |
| Fagus-Cilento | [1837] | [2007] | [171] | [0.41] | [0.26] | 7 | 1 | - | - | - | - | - | - | - |
| QCIBG*; *** | [1897] | [2013] | [117] | [0.60] | [0.66] | 9 | 0 | - | - | - | - | - | - | - |
| QFIMP1 | 1851 | 2013 | 163 | 0.50 | 0.78 | 34 | 0 | - | - | - | - | - | - | - |
| QFIMP2 | 1854 | 2013 | 160 | 0.55 | 0.79 | 34 | 0 | - | - | - | - | - | - | - |
| Fagus-Sila | [1854] | [2009] | [156] | [0.30] | [0.21] | 4 | 3 | - | - | - | - | - | - | - |
| Fagus-Parco-Aspromonte | [1874] | [2009] | [136] | [0.27] | [-0.42] | 5 | 2 | - | - | - | - | - | - | - |
| TOTAL | 1785**** | 1989**** | 205**** | 0.55**** | 0.80**** | 385 | 10 | 1750 | 1980 | 231 | 0.55 | 0.83 | 148 | 0 |
| | mean | mean | mean | mean r | mean EPS | sum (all sites) | sum (all sites) | mean | mean | mean | mean r | mean EPS | sum | sum (all sites) |

1  **Table 3:** Reconstruction statistics computed for both regressions and scaling over the inverted subperiods of calibration and
2  verification. RE = Reduction of error; CE = Coefficient of efficiency.

|  |  | $R^2$ | Regression | | Scaling | | Full period $R^2$ |
|---|---|---|---|---|---|---|---|
|  |  |  | RE | CE | RE | CE |  |
| Calib. | 1781-1880 | 0.383 |  |  |  |  |  |
| Verif | 1881-1980 |  | 0.484 | 0.305 | 0.533 | 0.371 |  |
| Calib. | 1881-1980 | 0.506 |  |  |  |  | 0.435 |
| Verif | 1781-1880 |  | 0.409 | 0.223 | 0.278 | 0.060 |  |

5

1 **Table 4:** Intercorrelation between reconstructed temperature series of late summer (AS; Trouet, 2014; Leonelli et al., this
2 study) and of summer (JAS; Klesse et al., 2015) based on tree-ring MXD in the study region. The correlation coefficients
3 were calculated over the common period 1714-1980, for both z-scores and 20 yr filtered series.
4

|  | AS Temp - TROUET_MXD | | AS Temp - LEONELLI_MXD_scaling | |
|---|---|---|---|---|
|  | z-scores | 20 yr gaussian | z-scores | 20 yr gaussian |
| **AS Temp** - LEONELLI_MXD_scaling | 0.85 | 0.74 | - | - |
| **JAS Temp** - KLESSE_MXD | 0.75 | 0.69 | 0.58 | 0.65 |

7
8

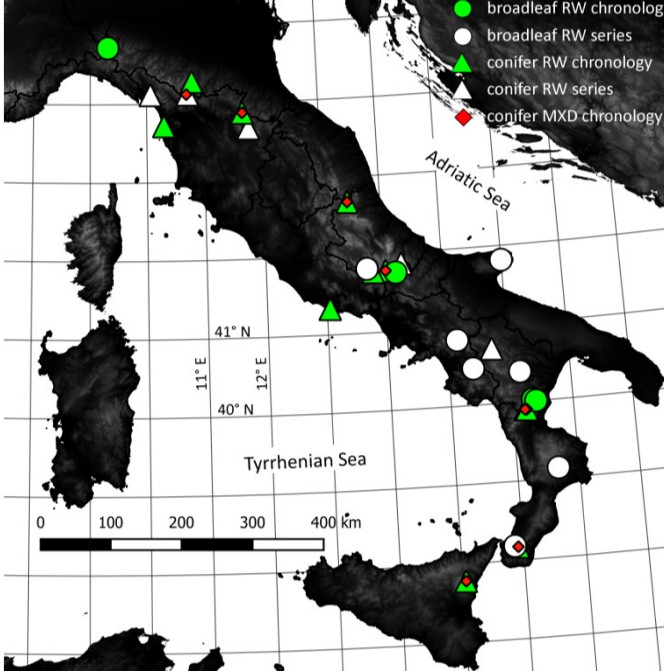

**Figure 1:** Distribution of the tree-ring sites from central and southern Italy available to the NEXTDATA project and used in this study. Sites were subdivided by the type of tree (conifer or broadleaf), the type of parameter (RW or MXD) and the type of data used (site chronology or only tree-ring series).

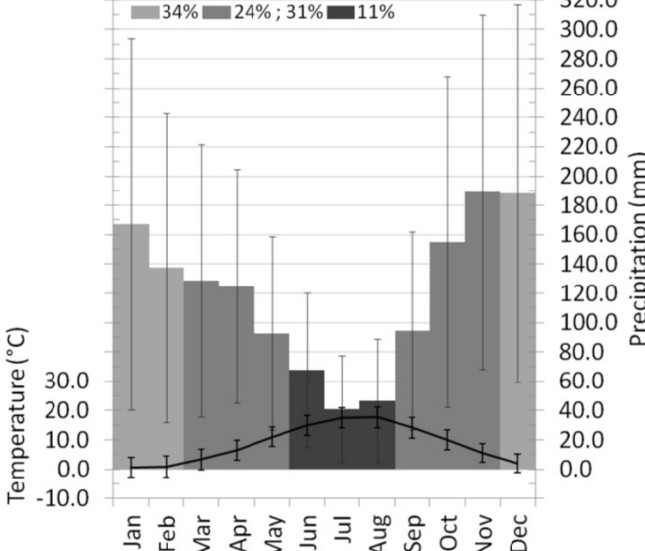

**Figure 2:** Monthly mean temperatures and precipitations over the period of 1880-2014 for all sites considered in this study. For both temperature and precipitation, the error bars indicate one standard deviation; for precipitation, the seasonal percentages of precipitation with respect to the mean annual value (= 1433 mm) are reported.

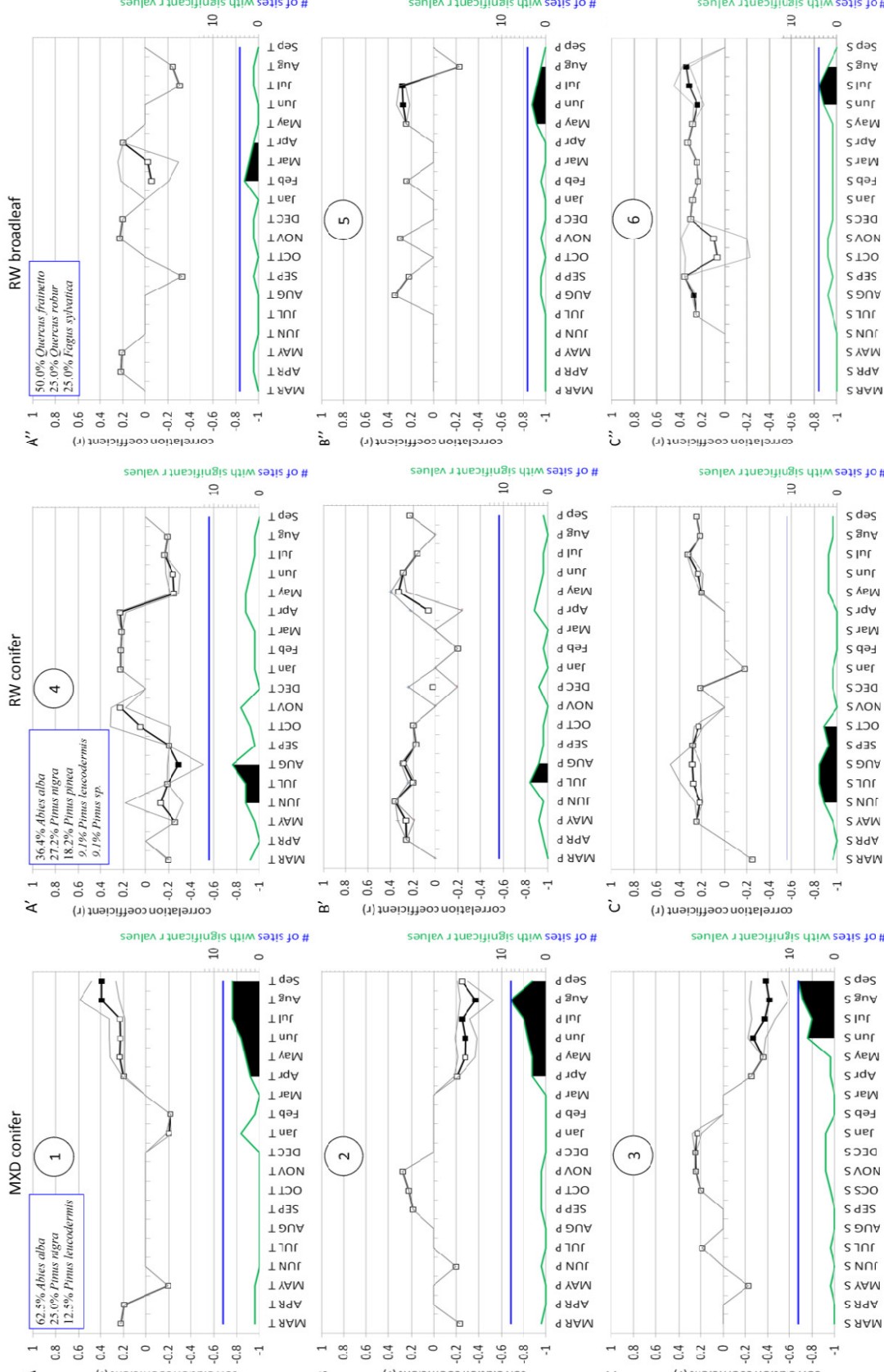

**Figure 3:** Bootstrapped correlation analysis performed over the common period of 1880-1980, considering chronologies of conifer MXD (left column; 3A, 3B and 3C), of conifer RW (center; 3A', 3B' and 3C') and of broadleaf RW (right; 3A", 3B" and 3C") vs. monthly temperature (A letters, first row), precipitation (B letters) and SPI_3 (C letters) from March of the year prior to growth to September of the year of growth. In A, A' and A" the percentages of the species composing the pool for each site used for the analysis is reported.

Means of statistically significant ($p<0.05$) correlation coefficient values (r) are depicted with squares, whereas maximum and minimum significant r values are indicated with grey lines; the blue lines depict the total number of sites in each comparison and the green lines indicate the total number of sites with statistically significant r values. Black-filled squares are given for those variables that show significant correlation values for at least 50 % of the total sites and have $|\bar{r}| > 0.25$; where both conditions occur, a circled number in the plot is given and the comparisons are selected for the following moving correlation analysis (Fig. 5). In each plot the climate variables with the highest number of sites with significant r values and nearby variables showing up to ¼ of this number are depicted with a black area.

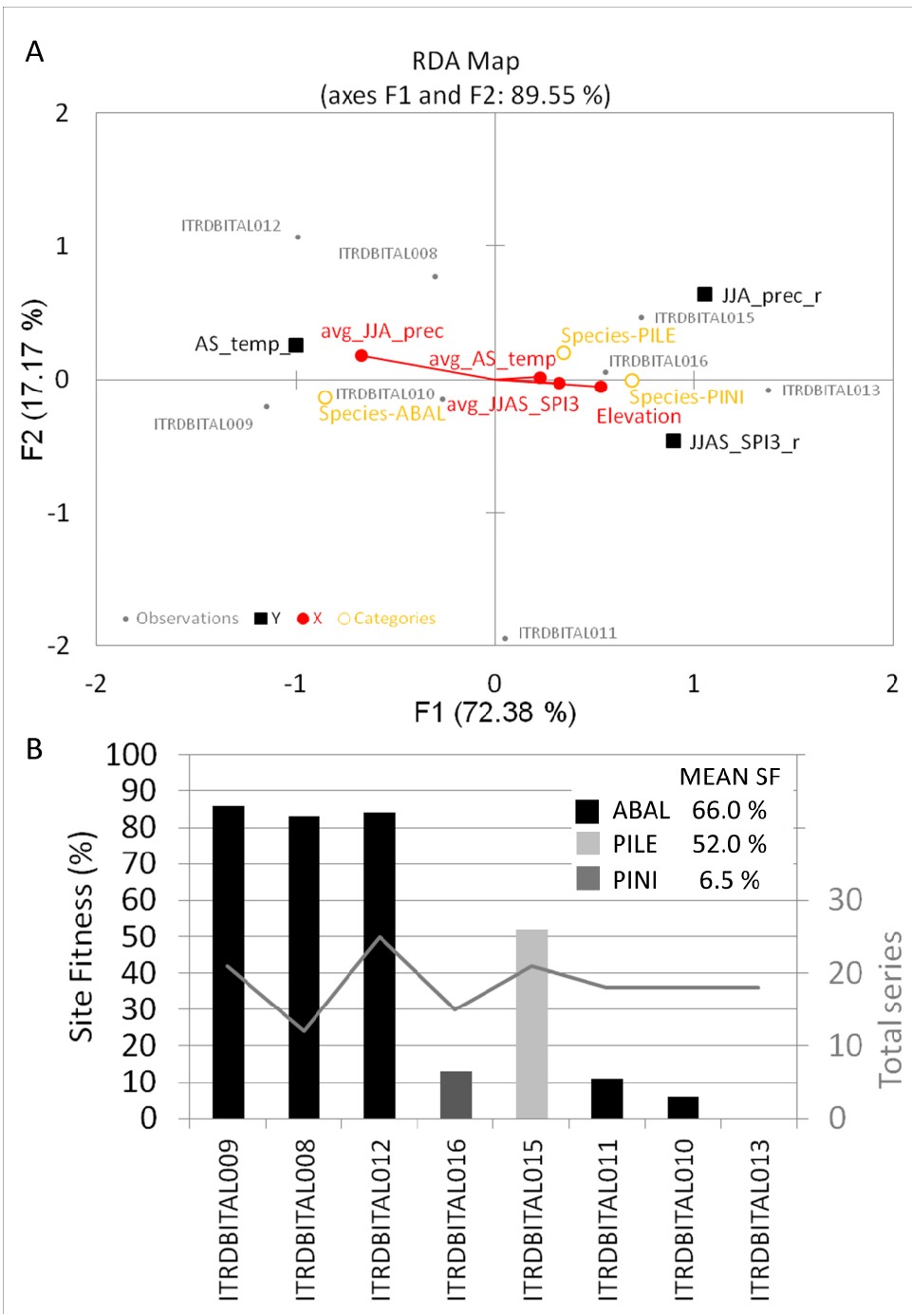

**Figure 4:** Ordination biplot (RDA analysis) of climate-growth relationships (response variables, Y) and environmental settings (explanatory variables X: elevation and climatic averages over the period 1880-1980) (4A). Site fitness evaluated on single indexed series included in the MXD HSTC chronology (SF; Leonelli et al., 2016) and total series per site (grey line) (4B). Sites are ordered with decreasing latitude along the x-axis. Mean SF values for each species are also reported. ABAL = *Abies alba*; PILE = *Pinus lucodermis*; PINI = *Pinus nigr*a.

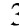

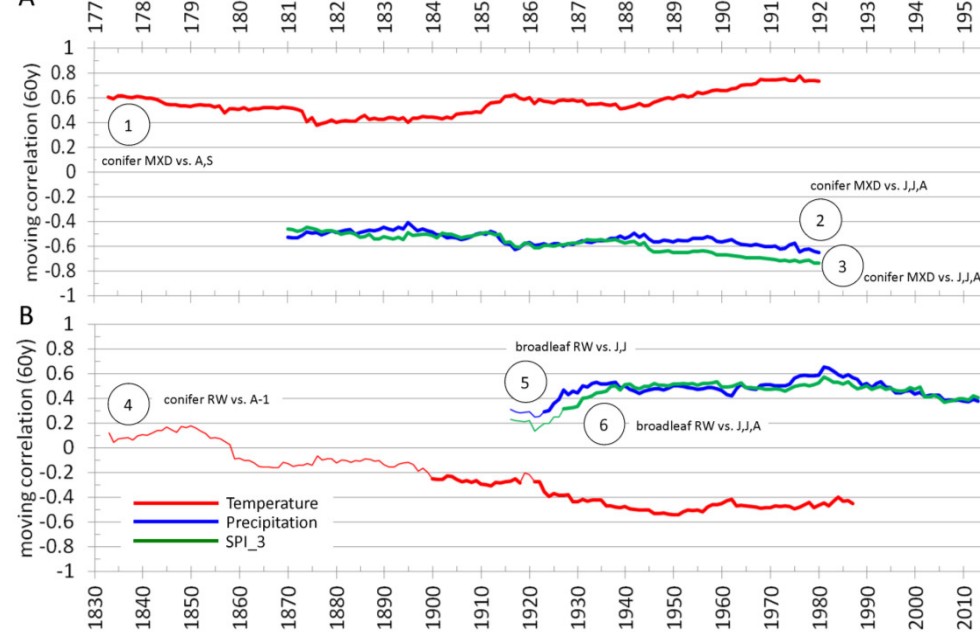

**Figure 5:** Bootstrapped moving correlation analysis with a 60 yr time window, performed over the maximum period available for the HSTC chronologies and their respective climate variables (temperature, precipitation and SPI_3) selected in the previous analysis (circled numbers as in Fig. 3). The statistically significant values (p<0.05) of r are depicted by bold lines.

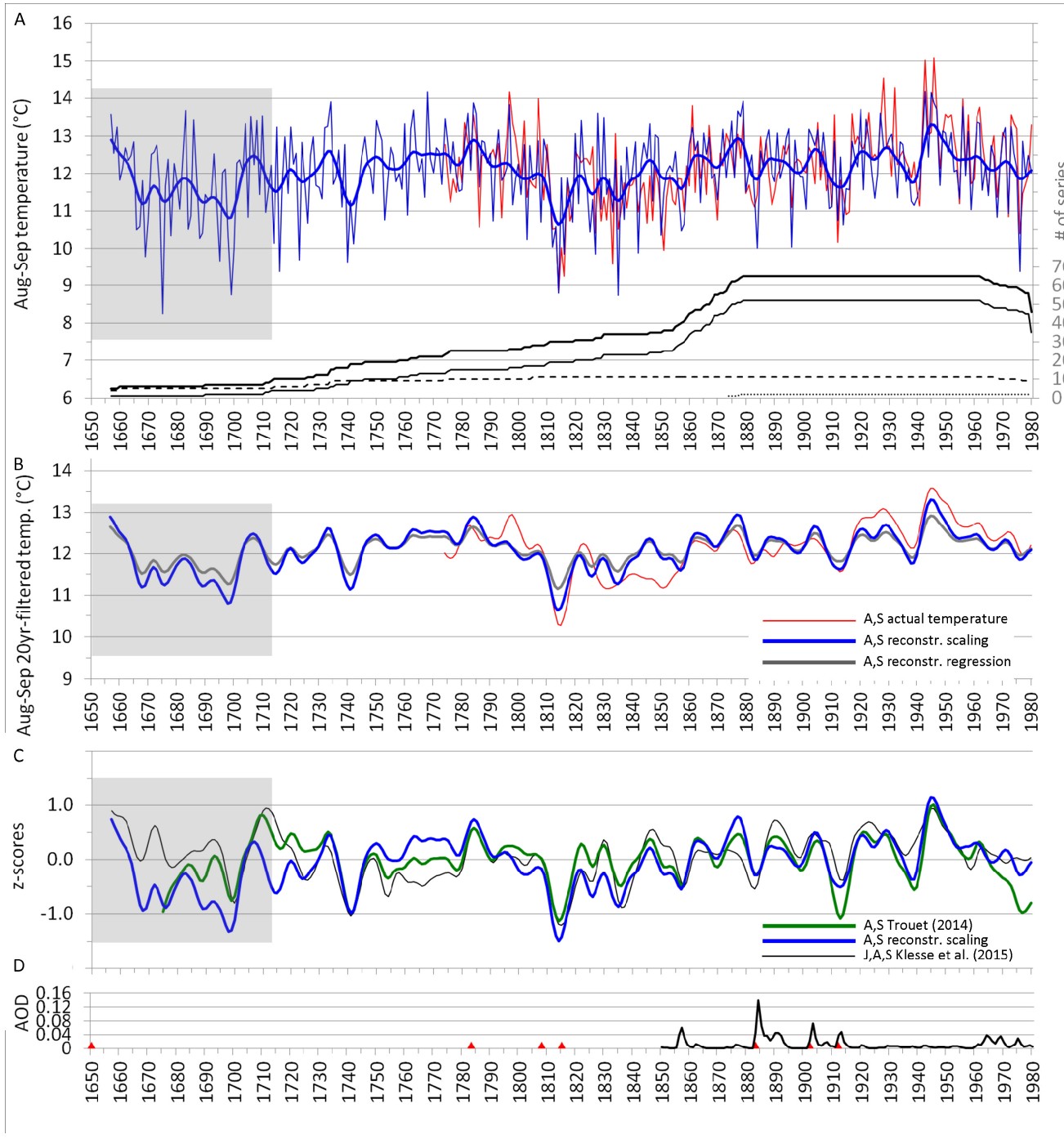

**Figure 6:** Reconstruction of late summer (August and September) temperature using the conifer MXD chronology with the scaling approach for the period 1650-1980 (6A). The bold black line indicates the total number of series (composed by a number of *Abies alba* (thin black line), *Pinus leucodermis* (dashed line) and *P. nigra* (dotted line) specimens). The low-pass filtered series with a 20 yr Gaussian smoother for both the reconstructions based on scaling and regression are also depicted (6B). The reconstructions were truncated when there was fewer than 5 trees, and the grey areas in the graphs depict the period where the conifer MXD chronology shows an EPS < 0.79 (prior to 1714, less than 10 trees. EPS > 0.85 since 1734). A comparison of the reconstructed late summer temperature (this paper) with the ones of Trouet (2014) and Klesse et al. (2015) using z-scores series (calculated over the common period 1714-1980 with EPS > 0.8 in all the original chronologies), filtered with a 20 yr Gaussian low-pass filter (6C). At the bottom the annual mean of stratospheric aerosol optical depth (AOD) at 550 nm for the Northern Hemisphere is reported (6D); dataset available at https://data.giss.nasa.gov/modelforce/strataer/ ; site accessed 2017-05-30; the red triangles mark major volcanic eruptions (Volcanic Explosivity Index ≥ 6): in chronological order Kolumbo-Santorini, Grímsvötn, Source unknown, Mount Tambora, Krakatau, Santa María, Novarupta.

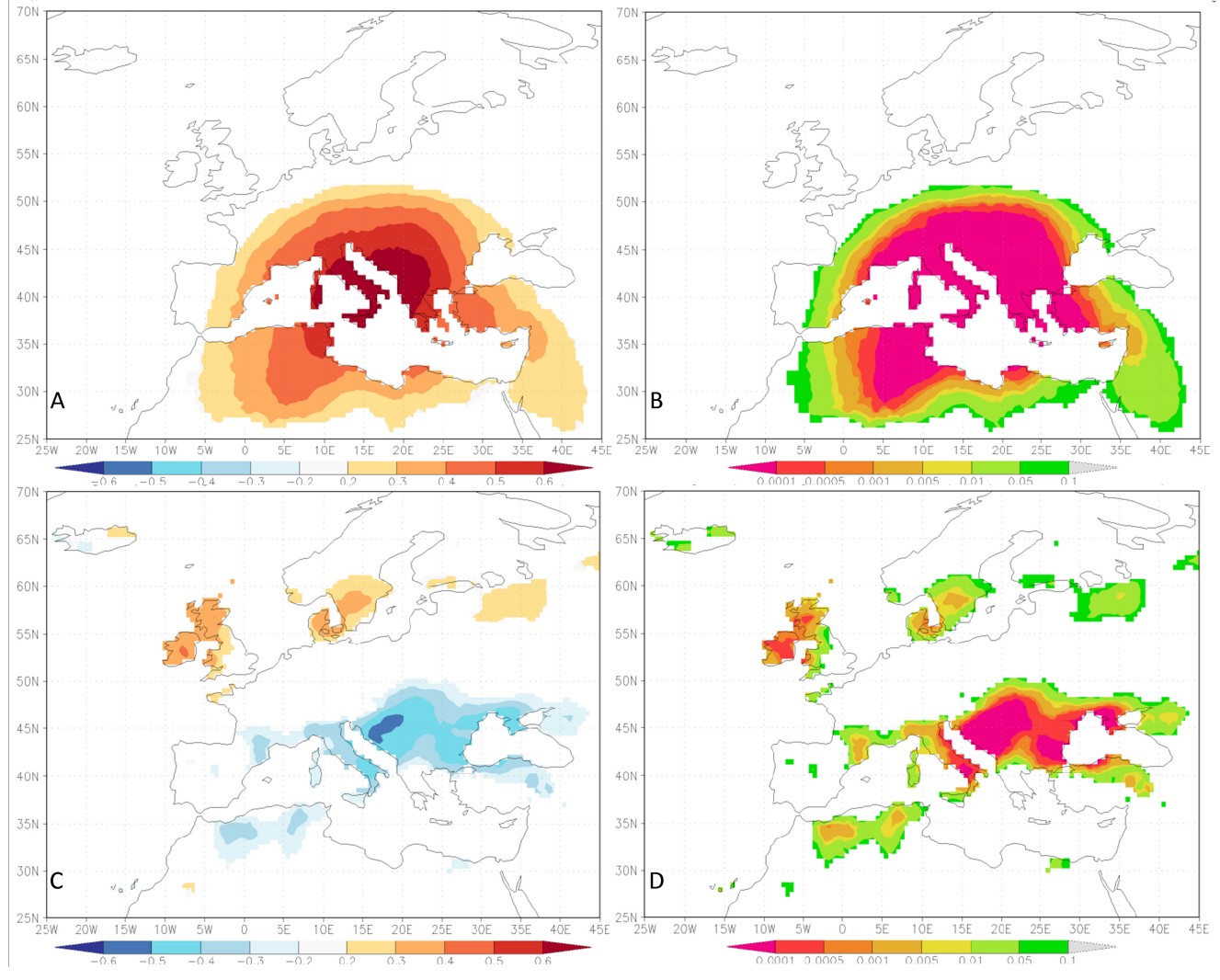

**Figure 7:** Spatial correlation pattern of the reconstructed late summer temperature (using the MXD chronology from the Italian Peninsula) versus the 0.5° grid CRU TS 4.0 August-September mean temperature (A, B) and mean precipitation (C, D), over the period of 1901-1980. Left side (A, C) Pearson's correlation coefficients, right side (B, D) the associated p values.