# Peer review of "Climate signals in a multispecies tree-ring network from central and southern Italy and reconstruction of the late summer temperatures since the early 1700s"

_Climate of the Past, 2017_

## Referee Comment (RC1) · Anonymous Referee #1 · 21 Mar 2017

Climate signals in a multi species tree-ring network...

This is an interesting and thoughtful, well written paper that I recommend should be accepted and published in Climate of the Past with minor revision. It describes the generation and analysis of a multi species tree-ring network from central and southern Italy, and a reconstruction of late summer temperatures since the early 1700s based on this network.

Using RW and MXD from 27 sites in Italy (both conifers and some hardwoods), temperature and precipitation signatures were identified, and eventually a late summer temperature reconstruction based on MXD was generated. There is apparent divergence between observed and reconstructed temperature of about 1 degree C, possibly due to the impact of drought stress.

Para beginning on line 8 of intro: good to reference some of recent modeling studies of subtropical drying due to climatic change in western North America, Mediterranean..(e,g. Seager et al. papers)

This paper has a good general overview/intro re the climate response in Mediterranean trees, which can be quite complex due to multiple influences on growth.

as found elsewhere, rather well behaved MXD temp signal, here linked to drought at high temperatures. would like to see more about the gradient of response to climate in these trees across space and elevation..

Would be good to discuss impact of climatic forcings on the region - e.g. the NAO (warm and cold season).. Also volcanic events - The year 1699 also seen as a cold year/interval elsewhere in Europe, North America following volcanism..

---

## Author Comment (AC1) · 29 Mar 2017

*Interactive comment* **- Anonymous Referee #1** - our responses in BLUE

Climate signals in a multi species tree-ring network...
This is an interesting and thoughtful, well written paper that I recommend should be accepted and published in Climate of the Past with minor revision. It describes the generation and analysis of a multi species tree-ring network from central and southern Italy, and a reconstruction of late summer temperatures since the early 1700s based on this network.
Using RW and MXD from 27 sites in Italy (both conifers and some hardwoods), temperature and precipitation signatures were identified, and eventually a late summer temperature reconstruction based on MXD was generated. There is apparent divergence between observed and reconstructed temperature of about 1 degree C, possibly due to the impact of drought stress.

Para beginning on line 8 of intro: good to reference some of recent modeling studies of subtropical drying due to climatic change in western North America, Mediterranean..(e,g. Seager et al. papers) This paper has a good general overview/intro re the climate response in Mediterranean trees, which can be quite complex due to multiple influences on growth.

Thank you for suggesting to expand the introduction including also the changes involving subtropical environments in general. In the reviewed version of the ms. we will add some reference to other researches dealing with this topic.

as found elsewhere, rather well behaved MXD temp signal, here linked to drought at high temperatures. would like to see more about the gradient of response to climate in these trees across space and elevation..

We agree with the Referee that a deeper analysis of climate/growth response across space and elevation would add some interesting information. Considering the strength of the signals recorded and the number of chronologies available, we focused our attention on the MXD chronologies, and performed a redundancy analysis (RDA) selecting as response variables the bootstrapped correlation coefficients of climate-growth relationships (Fig. 3) and as explanatory variables the environmental variables (geographical characteristics and climatic averages over the period 1880-1980). In order to attenuate co-variation within the environmental variables, a PCA was run before the RDA and the following variable were chosen:

- Elevation (co-varying with Longitude: our sites are placed at higher elevation at increasing longitude);
- avg_AS temperature;
- avg_JJA precipitation (co-varying with Latitude: higher latitude means higher precipitation amounts)
- avg_JJAS SPI3

From the analysis we obtained the following figure:

[Figure]

**Caption:** Ordination biplot *(RDA analysis) of climate-growth relationships (response variables, Y) and environmental variables (explanatory variables X: elevation and climatic averages over the period 1880-1980).* ABAL = *Abies alba*; PILE = *Pinus lucodermis*; PINI = *Pinus nigra.*

The strength of the AS temperature signal recorded in the MXD chronologies depends also on summer precipitation amounts (co-varying with Latitude, in our dataset of MXD) and Elevation (co-varying with Longitude, in our dataset of MXD): positive and negative correlation, respectively. Summer precipitation amounts and elevation correlate negatively in our dataset of MXD, underlining the prevalence of the latitudinal gradient of higher precipitation at north over the expected altitudinal gradient of higher precipitation at higher altitudes: sites at north, even if at lower altitudes, receive more summer precipitation than sites at south, at higher altitude. The RDA analysis shows that summer precipitation amounts and elevation are the most influencing the AS temperature signal: the F1 axis alone explains up to 72% of the variance in response variables, and especially in AS temperature and JJAS SPI3.
Of the considered explanatory variables, it is especially the latitudinal regime of summer precipitation amounts that modulates the sensitivity to AS temperature and to summer drought: sites at north (more mesic and at lower elevation) show stronger climate signals than sites at south (more xeric and at higher elevation). Even if southern sites are at higher elevation (and this is usually supposed to give trees a higher sensitivity to temperature) they are less sensitive to the selected climate variables.
Considering the responses related to the type of species, it is evident that in our dataset the influence of AS temperature on MXD, in *Abies alba* is more affected by summer precipitation amounts than in *Pinus lucodermis* and *P. nigra*. On the other hand, the influence of summer drought
on MXD, in pines is  more affected by elevation.
We will put this analysis in the online materials as the objective of explaining climate-growth
responses with respect to elevation or latitude is not a major objective of the paper. We will add
some considerations on climate/growth response across space and elevation in the Discussion
chapter.
Would be good to discuss impact of climatic forcings on the region - e.g. the NAO (warm and cold season).. Also volcanic events - The year
1699 also seen as a cold year/interval elsewhere in Europe, North America following volcanism..
In the reviewed version of the ms. we will add some considerations about the NAO and the volcanic
events in the Discussion chapter.
We thank the Referee for his help in critically reading the ms.
                                                                          *The Authors, March 29th, 2017*

---

## Referee Comment (RC2) · Anonymous Referee #2 · 2 May 2017

First of all, my apologies for delivering the review so late.

This paper gives a nice overview over climate responses in central-southern Italy across multiple species and reports a 300 year long late summer temperature record based on MXD. However, it is not really clear to me whether this paper tries to be a synthesis, a network analysis or about climate reconstruction, as neither part is performed sufficiently to justify publication in the present form. Had this been published in the 10-20 years ago the manuscript would have probably driven me to write a more positive review.

However, it's 2017 now and given the network size and actuality of the data I was actually wondering what is the added value of this publication over previous publications of Carrer et al. 2010, Piovesan et al. 2005 and Trouet 2014 apart from being the first simultaneous assessment of MXD/TRW and TRW of broadleaf/conifers? The former two of which have substantially higher site replication (Carrer et al. (2010) 55 ABAL sites and Piovesan et al. (2005) 24 FASY sites) and come up with very similar climate response patterns. Also, a big part of the manuscript is about climate reconstruction, solely based on conifer MXD data already published in Trouet 2014.

Trouet 2014 includes 6 of your 8 MXD chronologies in her Balkan temperature reconstruction, hence there is no surprise that the climate fingerprint is near to exactly the same. It's also no surprise that the temporal pattern is nearly the same. I also wouldn't say that the Trouet reconstruction is more variable in time, maybe on (multi-) decadal time-scale, but certainly not on centennial time-scale. Trouet 2014 varies around 0, whereas your chronology has a positive mean since 1850 and clearly negative before 1700. I would be interested to see actual statistics like standard deviation for such a claim (in low- and high-frequency domain), given the different amount of low frequency between your chronology is simply due to the different type of detrending used, which is discussed nowhere in the manuscript. As Klesse et al. 2015 use also RCS in Greece for an update of Mt. Olympus, a comparison of your data with completely independent data with potentially similar low-frequency characteristics is also lacking in this manuscript.

Was there no way to get Carrer and Piovesan/Di Filippo and others to provide their data to be included in this analysis? I know, Dendro people can be pretty possessive and restrictive with their data. But you cannot really call the present collection a representative network, any result is based on the screening of so little data (4 broadleaf chronologies; again, given that it's 2017 and not 2000) when there is potential for so much more. And even if the results kind of match previous publications, where is the novelty apart from applying the HSTC method?

A novelty would have been to tease apart the reasons for different strengths of climate influence, as you have done in your reply to Reviewer #1. That is what I would expect of a multi-species network analysis. The analysis and discussion presented in the manuscript is way too superficial. You could go much further and talk about which series from which sites, which species end up being highly sensitive? Is there a trend in mean climate conditions? And so on...

The authors furthermore exclude many of the in table 1 listed chronologies for the initial analysis, because they do not meet the criteria of number of samples or the required EPS threshold value. Later on, nowhere in the manuscript they state how many and which of the series in the HSTC approach come from the initially discarded sites, or which series of the initial good chronologies were discarded. Please indicate! How did you validate your site chronologies with only 3 series? Did you use other chronologies? If so, please specify in the manuscript!

Additionally, there are a couple of more chronologies on the ITRDB that fall into your region, uploaded in 2014 from P. Cherubini (your co-author). Did you exclude them because they were too short? If so, please specify in the manuscript!

Also you use RCS. How did you detrend the sites with less than 10 samples for the HSTC approach? There is no mention of it in the manuscript. And even 10 samples for a site RC is incredibly low. I am very skeptical about the use of RCS with such low replications as the ones used in the manuscript. Why didn't you just use a stiff spline detrending, or the classic negative exponential curve? What is the low-frequency gain over those approaches that are much less prone for weird sampling related trends (especially with low replication), since your chronologies are only (or >99%) composed of living material?

I challenge that the site-specific historical climatic records actually give you any real advantage over e.g. CRU, when you use correlation analysis (apart from the length of the record back to ~1800). Had you reported site-specific sensitivities, i.e. as regression slopes, to a parameter given a specific mean condition I would totally agree with you.

Temporal stabilities in climate correlations for ABAL and FASY TRW have been also reported previously (again, see Carrer et al. 2010 and Piovesan et al. 2008). So the only real novelty is the analysis with MXD. Is the correlation decay in conifer TRW due to opposing low-frequency trends (possibly related to your detrending) or is it the high-frequency agreement that decays? No discussion about that in the manuscript.

Furthermore the balance between Introduction/M&M and Results/Discussion is off. Especially the whole climate reconstruction section (1.2) takes an unreasonable large part of this manuscript. The main message could be condensed quite severely. If you insist on keeping it as detailed as possible then for the sake of completeness (as you seem to count every single recent climate reconstruction of the Mediterranean region) you should include as well: Dorado-Liñan et al 2015 (Spain, PINI, temp pJASO), Klesse et al. 2015 (Greece, PINI, MJJ precip; PILE, JAS temp), Levanic et al 2015 (Albania, PINI, JJ temp), Poljansek et al., 2013 (Bosnia-Herzegovina, PINI, summer sunshine), Tegel et al. 2014 (Albania, FASY, summer temp). All of which seem to me to have much more relevance to be cited than the chronologies from Turkey/Caucasus/Jordan, which come from far more distant locations (and in part use different species).

For the amount of different analyses performed, the result section is pretty short and the discussion in the context of previous publications in southern-central Italy again very superficial.

This manuscript needs some serious overhaul in its concept, structure and depth until it is acceptable for publication. M&M and Results have been written a lot in passive voice, which should be considered to be changed. Please use more active voice, as Word tells me directly to revise the previous sentence.

Some additional things:

Abstract Line 34: climate worsening is an awkward formulation, use climate cooling instead.

Table 2: # of series; be consistent in respect to reporting number of trees or cores. Or why are there only 11 and 15 series from Lombardi et al. 2008 (Co-author here) included? In that paper they report 25 and 30 series from those sites.

Figure 3: I suspect that rows A, B, C show the correlations with T, P and S, respectively? Please make that both clearer in the annotation and in the figure. Something like: "chronologies of conifer MXD (left), of conifer RW (center) and of broadleaf RW (right) vs. Monthly temperature (a), precipitation (b) and SPI_3 (c)".

Page 6, lines 27-31: What did you do exactly? The first two sentences don't make sense. You identified your DCV and z-scored this time-series? SPI is already z-scored. And why do you then retransform them, just leave them in the original unit if you use site-specific climate data.

And why didn't you use SPI-1 instead of monthly precipitation? Monthly precip is essentially SPI-1 before transforming the measured values into a gamma distribution and z-scoring based on the cumulative distribution, so the correlation changes only maybe at the second or third value after the point. This is nitpicking, but I was just wondering why you use both variables and don't decide for one of them.

---

## Referee Comment (RC3) · Anonymous Referee #2 · 11 May 2017

I apologize for "over-reading" the 100-y length condition and comments regarding this topic. Sorry to hear that people are still so uncooperative regarding sharing data that have been published >5-10 years ago. This sure is a problem for advancing the science and has been recognized (or better finally publicly "criticized") recently in Babst et al. 2017 (Improved tree-ring archives will support earth-system science. NEE).

Regarding RCS: Yes, for retaining low-frequency it's superior - given that your dataset actually allows a robust regional curve - but prone to a lot of biases. I am not really concerned about the MXD data, because the slope in MXD is usually pretty flat, so you

won't run in to big troubles there.

However, I would be still very interested to see to see the Italy-only MXD chronology detrended with a 150-year spline (if I remember correctly) for a direct comparison of the different oscillations against the Trouet reconstruction. I would consider the RCS application as a second and final step to investigate how much more low-frequency there actually is (or might be).

Additionally, I am still extremely cautious of the application of RCS on TRW at sites with n<10-15. If you use a "10% spline" (which in your case comes close to 15-20 years with the "younger" broadleaf samples) to build the RC, the RC is potentially very noisy (or wiggly). And if your 3-9 samples have a narrow age range you essentially take out most of the low frequencies you intended to retain and your RC at higher ages is probably more flexible at higher ages (due to only very few samples) than the stiff tail of a negative exponential curve.

Not giving an actual number, Esper et al. 2003 and Briffa & Melvin 2011 propose "the more samples the better", which between the lines is a minimum replication per year at 10 but coming from a population of >30 in total. Specifically Melvin (2004, Historical Growth Rates and Changing Climatic Sensitivity of Boreal Conifers, Section 6.3.3), stated you actually would need 62 samples per year for RCS to get the same per year standard deviation and confidence intervals as a 30-year spline chronology with n=10 (using Torneträsk and Finish-Lapland chronologies). "The cost for the inclusion of low-frequency variance is a requirement for greater tree replication in order to maintain similar confidence levels."

Although somewhat arbitrary it is common practice to set the EPS threshold to 0.85. The inclusion of EPS values down to 0.7 in your study tells me a lot about the "weak" coherence within your RCS chronologies (even the ones with "higher" replication of 16) during the common 1880-1980 interval. I assume the statistics would be higher (= more robust chronology) if you used a stiffer spline (∼150 years) or negexp detrending

instead. What are the statistics for the final RCS-HSTC-chronologies, are they >0.85?

---

## Author Response (AR1)

Dear Prof. Luterbacher,

We have changed the previous version of the ms. responding to all points of concern, following the Reviewers' suggestions and yours.

In particular, in order to have a lighter method chapter (as suggested by Referee#2), shortened the text on the construction of climate series, being not the focus of the current paper. The introduction was slightly shortened here and there (as suggested by Referee#2), but we added some parts on 'subtropical drying due to climatic change in western North America, Mediterranean' (as suggested by Referee#1).

Moreover, we compared our reconstruction of AS temperature also with the JAS reconstruction of Klesse et al. (2015) based on one MXD chronology from Mt. Olympus (Greece), as suggested by Referee#2 and with the dataset of JJA temperature of Luterbacher et al. (2004) available on Climate Explorer, as suggested by you. The comparison between tree-ring based temperature is now performed in Table 4 and in the new Fig. 6C that was also enhanced with information on past volcanic eruptions (Referee#1) and mean hemispheric values of incoming solar radiation. The comparisons between Luterbacher et al. (2004) JJA temperature and the tree-ring based reconstructions of AS (North Eastern Mediterranean and central and southern Italy) and JAS (Greece) temperature is presented in the Online Material 1, as it concerns different variables and different datasets from different territories. It is performed considering also JJA, JAS and As temperature records from Brumetti et al. (2006), whereas we do not use the spring and autumn gridded dataset of Xoplaki et al. (2005) and the precipitation gridded dataset of Pauling et al. (2006), as our dataset of AS, is more linked to summer temperature.

We also added a new figure (Fig. 4), in order to give more insights about climate-density responses at the site level and about the percentage of HSTC trees per site, according to site location and species (both Referees asked for deeper details on these topics).

Hereafter you can find an updated point-to-point response to the received comments.

We thank the Referees and you for critically reading the ms. We hope that the current version of the ms. has reached the requested quality standard of CP. In case of acceptance, the data will be made available as Online Material 2.

Kind regards,
Giovanni Leonelli and authors

*our updated* responses in BLUE

Climate signals in a multi species tree-ring network...
This is an interesting and thoughtful, well written paper that I recommend should be accepted and published in Climate of the Past with minor revision. It describes the generation and analysis of a multi species tree-ring network from central and southern Italy, and a reconstruction of late summer temperatures since the early 1700s based on this network.
Using RW and MXD from 27 sites in Italy (both conifers and some hardwoods), temperature and precipitation signatures were identified, and eventually a late summer temperature reconstruction based on MXD was generated. There is apparent divergence between observed and reconstructed temperature of about 1 degree C, possibly due to the impact of drought stress.

Para beginning on line 8 of intro: good to reference some of recent modeling studies of subtropical drying due to climatic change in western North America, Mediterranean..(e,g. Seager et al. papers) This paper has a good general overview/intro re the climate response in Mediterranean trees, which can be quite complex due to multiple influences on growth.
As requested by Referee#1, we added in the Introduction a paragraph on researches dealing with the impacts of hydroclimatic changes on subtropical environments of North America and the Mediterranean: Fu et al., 2006; Seager et al., 2007; Seager and Vecchi, 2010; Schlaepfer et al., 2017. Page 2, Lines 10-15.

 as found elsewhere, rather well behaved MXD temp signal, here linked to drought at high temperatures. would like to see more about the gradient of response to climate in these trees across space and elevation..
We agree with the Referee that a deeper analysis of climate/growth response across space and elevation would add some interesting information. Considering the strength of the signals recorded and the number of chronologies available, we focused our attention on the MXD chronologies, and performed a redundancy analysis (RDA) selecting as response variables the bootstrapped correlation coefficients of climate-growth relationships (Fig. 3) and as explanatory variables the environmental variables (geographical characteristics and climatic averages over the period 1880-1980). In order to attenuate co-variation within the environmental variables, a PCA was run before the RDA and the following variable were chosen:

- Elevation (co-varying with Longitude: our sites are placed at higher elevation at increasing longitude);
- avg_AS temperature;
- avg_JJA precipitation (co-varying with Latitude: higher latitude means higher precipitation amounts)
- avg_JJAS SPI3

See Method chapter, Page 6, lines 11-22.

From the analysis we obtained the following figure that is now included in the ms. as Fig. 4A:

[Figure]

**Figure 4A:** Ordination biplot *(RDA analysis) of climate-growth relationships (response variables, Y) and environmental variables (explanatory variables X: elevation and climatic averages over the period 1880-1980).* ABAL =*Abies alba*; PILE = *Pinus lucodermis*; PINI = *Pinus nigra*.

The Results chapter was updated, page 8 lines 1-12: The strength of the AS temperature signal recorded in the MXD chronologies depends also on summer precipitation amounts (co-varying with Latitude, in our dataset of MXD) and Elevation (co-varying with Longitude, in our dataset of MXD): positive and negative correlation, respectively.

Summer precipitation amounts and elevation correlate negatively in our dataset of MXD, underlining the prevalence of the latitudinal gradient of higher precipitation at north over the expected altitudinal gradient of higher precipitation at higher altitudes: sites at north, even if at lower altitudes, receive more summer precipitation than sites at south, at higher altitude. The RDA analysis shows that summer precipitation amounts and elevation are the variables most influencing the MXD sensitivity to AS temperature: the F1 axis alone explains up to 72% of the variance in response variables, and especially in AS temperature and JJAS SPI3 signals.

These results and some considerations on climate/growth response across space and elevation were added also in the Discussion chapter, Page 9 line 19-27:
Of the considered explanatory environmental variables, it is especially the latitudinal regime of summer precipitation amounts that modulates the MXD sensitivity to AS temperature and to summer drought (Fig. 4A): sites at north (more mesic and at lower elevation) show stronger climate signals than sites at south (more xeric and at higher elevation). Even if southern sites are at higher elevation (and this is usually supposed to give trees a higher sensitivity to temperature) they are less sensitive to the selected climate variables. MXD sites from southern Italy present a markedly lower SF than sites from central-northern Apennines. Considering the responses related to the type of species, it is evident that in our dataset the influence of AS temperature on MXD, in Abies alba is more affected by summer precipitation amounts than in Pinus lucodermis and P. nigra. On the other hand, the influence of summer drought on MXD, in pines is more affected by elevation.

Would be good to discuss impact of climatic forcings on the region - e.g. the NAO (warm and cold season).. Also volcanic events - The year 1699 also seen as a cold year/interval elsewhere in Europe, North America following volcanism..
We added some considerations about the NAO and the volcanic events in the Discussion chapter, namely from Page 10 line 43 to page 11 line 7:
Tree-ring growth may be affected also by large-scale climate variability, such as the North Atlantic Oscillation (NAO), the prominent mode of atmospheric circulation in the North Atlantic that affects temperature and precipitation patterns in Europe (D'Arrigo et al., 1993; Cook et al., 2002). In the eastern Mediterranean region, a teleconnection with summer climate conditions in the British Isles has been found in a summer temperature reconstruction for Bulgaria (Trouet et al., 2012), where tree-ring growth patterns are strongly linked to drought conditions. For Greece and the region eastward (Klesse et al., 2015), a prominent dipole pattern of summer NAO was found, whereas in Italy a major effect on tree growth was found for winter NAO, that correlates negatively with winter precipitation amounts, responsible of soil moisture during the growing season (Piovesan and Schirone, 2000).

The impacts of large volcanic eruptions on MXD was discussed from Page 9 line 37 to page 10 line 9: An important factor influencing the tree-ring MXD is volcanism, especially in correspondence of highly explosive eruptions that can change the intensity of the incoming solar radiation and that are able to change circulation patterns and to cool the climate at hemispheric to global scale (e.g., Briffa et al., 1998). The largest explosive eruptions (Volcanic Explosivity Index $\geq$ 6; Siebert et al., 2011) correspond to local minimum densities in the tree rings (Fig. 6C and 6D), and some of them are well known years of famine and low crop yields. The year 1699 and the proceeding decade is known for being related to recurrent explosive eruptions in Iceland and Indonesia (Le Roy Ladurie, 2004), inducing great famines around Europe and North America (Mitchison, 2002). The 1809 eruption of source unknown (Guevara-Murua et al., 2014) and the 1815 eruption of Mount Tambora induced a decade of very low summer temperature and high precipitation (Luterbacher and Pfister, 2015). This was the coldest decade of the so called Little Ice Age (Lamb, 1995), corresponding also to glacier advance phases in the Alpine glaciers, that reached their first maximum extent of the Holocene (the second and last, being around 1850; e.g., Matthews and Briffa, 2005). In 1883 Mount Krakatoa and in 1914 Mount Pinatubo eruptions correspond to local minima in the MXD. But a straightforward correspondence between minimum values of MXD densities and large eruption is lacking: some differences at the regional scale with respect to global scale may occur due to local circulation patterns or the presence of seas, as it is the case of the 1783 Grímsvötn Volcano eruption (Iceland), that corresponds to unexpected high MXD densities in the tree rings from the Mediterranean area (Fig. 6) but not at the global scale (see Fig. 1 in Briffa et al., 1998), or the local minimums of MXD density of 1740 and 1938 found in this paper that are not linked to any particular large eruption.

Volcanic eruption were also detailed as possible environmental factors locally masking the climatic signals in the tree-ring chronologies, Page 11 lines 12-14: ...may impact tree-ring growth as well as the presence of an active volcano and its direct influence on local climate and atmospheric conditions (such as the Vesuvio Volcano, Battipaglia et al., 2007, or the Etna Volcano, Sailer et al., 2017).

Information on major volcanic eruptions (Volcanic Explosivity Index $\geq$ 6) and the mean hemispheric values of incoming solar radiation through time was also added in Fig. 6D.

Interactive comment **- Anonymous Referee #2 -**
our *updated* responses in BLUE

This paper gives a nice overview over climate responses in central-southern Italy across multiple species and reports a 300 year long late summer temperature record based on MXD. However, it is not really clear to me whether this paper tries to be a synthesis, a network analysis or about climate reconstruction, as neither part is performed sufficiently to justify publication in the present form. Had this been published in the 10-20 years ago the manuscript would have probably driven me to write a more positive review.

The paper presents only original results, some are innovative some others are a confirmation of what already found, fact that underlines the goodness of the applied methods on the available dataset. Of course we strived for improving the ms. quality during the constructive review process. We compiled a cleaned large-scale network of long (> 100 yr) tree-ring chronologies from the Italian Peninsula to identify signals of climate variability in indices of tree growth, for a climate-change vulnerable region. The applied methodology is the key, innovative, issue of our paper.

Specifically, the paper presents the application for the Italian peninsula of an innovative approach to climate reconstruction, firstly approved in the dendro community in 2016 (i.e., 1 year ago; Climatic Change, 2016, 137:275–291, DOI 10.1007/s10584-016-1658-5), but the climate reconstruction is not the only objective: a deep analysis of climate signals recorded by trees (RW and MXD) in a regional-scale network is performed on static periods (using site chronologies, classical approach) as well as on moving periods (using HSTC chronologies, innovative approach) in order to evaluate reconstruction potentials and possible biases in past climate reconstructions.
Briefly, rather strict passages of quality check of each individual series vs. the respective mean chronology are performed before constructing the site chronology with dendroclimatic purposes (only older than 100 yr trees, etc. ; p. 6 l. 13 and following lines). Not all the resulting site chronologies are used (see the problematic gray-shaded areas in Table 2), and for these latter sites only the individual indexed series are retained for further analyses. Finally all the 'saved' individual indexed series from all sites are initially used for the construction of the HSTC chronology (p. 6 l. 32 and following).
To our knowledge this is the first attempt performed in the Italian peninsula presenting a multispecies and multiproxy approach with dendroclimatic purposes.

However, it's 2017 now and given the network size and actuality of the data I was actually wondering what is the added value of this publication over previous publications of Carrer et al. 2010, Piovesan et al. 2005 and Trouet 2014 apart from being the first simultaneous assessment of MXD/TRW and TRW of broadleaf/conifers? The former two of which have substantially higher site replication (Carrer et al. (2010) 55 ABAL sites and Piovesan et al. (2005) 24 FASY sites) and come up with very similar climate response patterns.
The added value for 2017 are the first application of the HSTC approach at the regional scale in the Italian peninsula, and the previous passages for the site chronologies construction; the simultaneous assessment of MXD/TRW and TRW of broadleaf/conifers, as also recognized by Referee2; the use of high quality site specific climate data, and the length of the meteorological series that has allowed us to calibrate and validate the models on 100 yr periods. Some other added values are hereafter reported.

Also, a big part of the manuscript is about climate reconstruction, solely based on conifer MXD data already published in Trouet 2014.
Trouet 2014 includes 6 of your 8 MXD chronologies in her Balkan temperature reconstruction, hence there is no surprise that the climate fingerprint is near to exactly the same. It's also no surprise that the temporal pattern is nearly the same.
Our reconstruction is performed using a different methodology than Trouet 2014, and is based only on the Italian sites, thus excluding surrounding areas characterized by more continental climates (i.e., the European Alps, Balkan area, Greece and sites from the central and eastern European Alps to central Romania and Bulgaria; p. 4 l. 20). Our reconstruction improves the one of Trouet 2014, being more representative for Italy and presenting less negative oscillations, especially in the recent period (see next heading).

I also wouldn't say that the Trouet reconstruction is more variable in time, maybe on (multi-) decadal time-scale, but certainly not on centennial time-scale.

Trouet 2014 varies around 0, whereas your chronology has a positive mean since 1850 and clearly negative before 1700. I would be interested to see actual statistics like standard deviation for such a claim (in low- and high-frequency domain), given the different amount of low frequency between your chronology is simply due to the different type of detrending used, which is discussed nowhere in the manuscript. As Klesse et al. 2015 use also RCS in Greece for an update of Mt. Olympus, a comparison of your data with completely independent data with potentially similar low-frequency characteristics is also lacking in this manuscript.

In the revised version we have added more information and comparisons with other reconstructions (namely Klesse et al., 2015) as suggested by Referee2, and with gridded climate series, as suggested by the Editor.

Around 1913 and in the 1970s the reconstruction of Trouet 2014 shows temperature nearly as low as during the coolest periods at the end of the Little Ice Age (around 1815), which is questionable or only partially explainable with her decision of including also sites characterized by continental climates. Our reconstruction is much less variable over the same periods. Three years after Trouet 2014, we are able to improve the reconstruction of late summer temperature for the region of the Italian peninsula (this is another added value).

Regarding the detrending method applied for the HSTC chronologies, this is widely presented in the ms. (p. 6 l. 32 up to l. 42): the HSTC are constructed starting from the indexed individual series that are obtained while applying the RCS method at each site (p. 6 l. 5 and following).

Was there no way to get Carrer and Piovesan/Di Filippo and others to provide their data to be included in this analysis? I know, Dendro people can be pretty possessive and restrictive with their data. But you cannot really call the present collection a representative network, any result is based on the screening of so little data (4 broadleaf chronologies; again, given that it's 2017 and not 2000) when there is potential for so much more. And even if the results kind of match previous publications, where is the novelty apart from applying the HSTC method?

If data is published is available to the community, otherwise it is not. The resulting dataset used and presented in the paper is what we could collect and, based on the innovative methods applied, on the high quality of the site specific climate data, and on the obtained results, we think that it is adequate for a publication in 2017.

A novelty would have been to tease apart the reasons for different strengths of climate influence, as you have done in your reply to Reviewer #1. That is what I would expect of a multi-species network analysis. The analysis and discussion presented in the manuscript is way too superficial. You could go much further and talk about which series from which sites, which species end up being highly sensitive? Is there a trend in mean climate conditions? And so on...

Based on this suggestion, we have added the RDA analysis on the dependence of climate-growth relationships on site settings and mean climate conditions (Fig. 4A). Moreover, we have add some more information as the ones suggested by Referee#2, focusing on the MXD series, the only one used for performing the climate reconstruction (Fig. 4B).

The authors furthermore exclude many of the in table 1 listed chronologies for the initial analysis, because they do not meet the criteria of number of samples or the required EPS threshold value. Later on, nowhere in the manuscript they state how many and which of the series in the HSTC approach come from the initially discarded sites, or which series of the initial good chronologies were discarded. Please indicate!

We added a new figure about Site Fitness at the MXD sites included in this paper. SF was calculated as the percentage of selected HSTC series of conifer MXD with respect to the total of series available at each site (Fig. 4B).

How did you validate your site chronologies with only 3 series? Did you use other chronologies? If so, please specify in the manuscript!

Site chronologies were all validated starting initially considering the whole dataset of individual raw series available. The final number of series per site is the result of the iterative selection applied to the initial datasets: we only retained series responding to the fixed criteria (p. 6 l 13 and following). No chronology from sites with a so low number of series entered in the successive analyses.

Additionally, there are a couple of more chronologies on the ITRDB that fall into your region, uploaded in 2014 from P. Cherubini (your co-author). Did you exclude them because they were too short? If so, please specify in the manuscript!

Our research is based on data available to the authors and to the dendro community in 2015 (year of last dataset update; this information has been added in the revised version of the ms.).
Referee#2 will agree with us that the chronologies uploaded onto the NOAA's ITRDB are data collected for many different research objectives, not only for investigating climate responses or for performing climate reconstructions. With our robust approach for chronology construction (deeply detailed in the method chapter p. 6 l. 13 and following lines), we had to discard several Italian sites, but there is no reason to make a list of the discarded sites and the reasons why they were discarded since the beginning: they simply did not meet all the requirements fixed by us for dendroclimatic analysis (most of the times they presented too short chronologies). The Cherubini's chronologies that the Referee#2 is mentioning were also checked.

Also you use RCS. How did you detrend the sites with less than 10 samples for the HSTC approach? There is no mention of it in the manuscript. And even 10 samples for a site RC is incredibly low. I am very skeptical about the use of RCS with such low replications as the ones used in the manuscript. Why didn't you just use a stiff spline detrending, or the classic negative exponential curve? What is the low-frequency gain over those approaches that are much less prone for weird sampling related trends (especially with low replication), since your chronologies are only (or >99%) composed of living material?

RCS is a well approved approach for retaining low-frequency variability in tree-ring chronologies (especially the long ones), and performs better than splines and negative exponentials in this domain. Our approach was to apply the same detrending method at each site and to the whole dataset, in order to treat all data in the same way. Sites with low replication presented however long and well intercorrelating individual series: if the resulting chronology presented high values of EPS then we used it in the following climate-growth analysis, otherwise we used only their indexed individual series.

I challenge that the site-specific historical climatic records actually give you any real advantage over e.g. CRU, when you use correlation analysis (apart from the length of the record back to ~1800). Had you reported site-specific sensitivities, i.e. as regression slopes, to a parameter given a specific mean condition I would totally agree with you.

The climate data used in this research are site-specific (coordinates, elevation and slope orientation), better homogenized and based on more stations than the ones used for the CRU gridded data. Most of the stations for central and southern Italy used for the CRU dataset start after 1950 and, before this date, the CRU interpolation scheme imports information from very far. We used the CRU as independent dataset for evaluating the spatial correlation pattern of our reconstruction (Fig. 7).
The use of long dataset has let us perform model calibration and verification on long time periods of 100 yr each.
In order to provide information on site specific and species specific responses of MXD to climate, we included in the ms the RDA analysis and the analysis of site fitness (new Fig. 4A and Fig. 4B). Please refer to the responses to Referee#1 (pages 2 and 3 this file).

Temporal stabilities in climate correlations for ABAL and FASY TRW have been also reported previously (again, see Carrer et al. 2010 and Piovesan et al. 2008). So the only real novelty is the analysis with MXD. Is the correlation decay in conifer TRW due to opposing low-frequency trends (possibly related to your detrending) or is it the high-frequency agreement that decays? No discussion about that in the manuscript.

Temporal stability in climate correlations was tested on HSTC chronologies of RW (broadleaf and conifer) and MXD (conifer), innovative aspect, and not on species. This analysis was only performed for evaluating the reconstruction potentials and the possible biases in past climate reconstructions. Some consideration on temporal stability of the signals recorded in conifer RW were discussed page 9 lines 35-37, always in the view of a potential climatic reconstruction.

Furthermore the balance between Introduction/M&M and Results/Discussion is off. Especially the whole climate reconstruction section (1.2) takes an unreasonable large part of this manuscript. The main message could be condensed quite severely. If you insist on keeping it as detailed as possible then for the sake of completeness (as you seem to count every single recent climate reconstruction of the Mediterranean region) you should include as well: Dorado-Liñan et al 2015 (Spain, PINI, temp pJASO), Klesse et al. 2015 (Greece, PINI, MJJ precip; PILE, JAS temp), Levanic et al 2015 (Albania, PINI, JJ temp), Poljansek et al., 2013 (Bosnia-Herzegovina, PINI, summer sunshine), Tegel et al. 2014 (Albania, FASY, summer temp). All of which seem to me to have much more relevance to be cited than the chronologies from Turkey/Caucasus/Jordan, which come from far more distant locations (and in part use different species). For the amount of different analyses performed, the result section is pretty short and the discussion in the context of previous publications in southern-central Italy again very superficial.

We thank Refere2 for the helpful suggestions, some of the suggested references were added. A more balanced ms. is here proposed; we shortened some parts dealing with the construction of the climate series and we focused more on the Mediterranean regions closer to our study area, following some of Referee2's suggestions. Climate-growth publications from Italy and the Mediterranean were mainly focused on the species used in our ms. (p. 2 l. 32).

This manuscript needs some serious overhaul in its concept, structure and depth until it is acceptable for publication. M&M and Results have been written a lot in passive voice, which should be considered to be changed. Please use more active voice, as Word tells me directly to revise the previous sentence.

We have performed many changes in the ms. as here reported in this file, and we have changed most of the text from passive to active voice, as suggested-

Some additional things:

Abstract Line 34: climate worsening is an awkward formulation, use climate cooling instead.

Probably climate cooling and/or wetter conditions as MXD has proven to depend both on temperature and precipitation/drought (Fig. 3). These variables, especially in summer, are also associated in Mediterranean climates (p. 9 l. 1). We have reworded the sentence, and along the ms we now report the years of minimum in the original series of temperature reconstruction: i.e., 1699, 1740, 1814, 1914, 1938).

Table 2: # of series; be consistent in respect to reporting number of trees or cores. Or why are there only 11 and 15 series from Lombardi et al. 2008 (Co-author here) included? In that paper they report 25 and 30 series from those sites.

Given the series selection method set up for this research, at each site some series were discarded if not meeting the fixed requirements (p. 6 l. 13 and following lines).

Figure 3: I suspect that rows A, B, C show the correlations with T, P and S, respectively? Please make that both clearer in the annotation and in the figure. Something like: "chronologies of conifer MXD (left), of conifer RW (center) and of broadleaf RW (right) vs. Monthly temperature (a), precipitation (b) and SPI_3 (c)".

Caption modified. The figure order is correct.

Page 6, lines 27-31: What did you do exactly? The first two sentences don't make sense. You identified your DCV and z-scored this time-series? SPI is already z-scored. And why do you then retransform them, just leave them in the original unit if you use site-specific climate data.

The sentence was simplified. Series were transformed in z-scores before averaging them between sites. The 'interesting' climate variables identified by black-filled squares in Fig. 3 (months with significant correlations at most sites (>50 %) and with mean correlation values of $|r| > 0.25$) were regionalized and then averaged over two to four consecutive months (we called them DCV). Regional climate series were calculated by z-scoring the monthly series and calculating regional mean departures; the series were then completed and ri-converted in original units (based on regional mean departures and their specific means and standard deviations), and finally averaged between sites. DCVs were then calculated as means of consecutive months of the regional series.

And why didn't you use SPI-1 instead of monthly precipitation? Monthly precip is essentially SPI-1 before transforming the measured values into a gamma distribution and z-scoring based on the cumulative distribution, so the correlation changes only maybe at the second or third value after the point. This is nitpicking, but I was just wondering why you use both variables and don't decide for one of them.

We actually used the SPI calculated at several timescales (from 1 to 12 months; p. 5 l. 36) when assessing climate-growth relationships. As explained in the Results chapter (p. 7 l. 32) 'the highest correlations (for both MXD and RW) were obtained for the indices calculated at the timescales of 2 and mainly of 3 months'. We therefore decided to present only the SPI_3 results, and this is also discussed later (p. 9 l. 4). This timescale is used for modeling agricultural droughts and well fits with growth and wood density issues also in trees. We prefer leaving in the ms. also the variable of precipitation, being it of more direct readability.

Interactive comment **- Anonymous Referee #2** -
our *updated* responses in BLUE
I apologize for "over-reading" the 100-y length condition and comments regarding this topic. Sorry to
hear that people are still so uncooperative regarding sharing data that have been published >5-10 years
ago. This sure is a problem for advancing the science and has been recognized (or better finally publicly
"criticized") recently in Babst et al. 2017 (Improved tree-ring archives will support earth-system
science. NEE).
With our effort of setting up a cooperative and open access dendro group in Italy, we will give the
opportunity to freely access our data in the Online Material 2.
Regarding RCS: Yes, for retaining low-frequency it's superior - given that your dataset actually allows a
robust regional curve - but prone to a lot of biases. I am not really concerned about the MXD data,
because the slope in MXD is usually pretty flat, so you won't run in to big troubles there.
However, I would be still very interested to see to see the Italy-only MXD chronology detrended with a
150-year spline (if I remember correctly) for a direct comparison of the different oscillations against the
Trouet reconstruction. I would consider the RCS application as a second and final step to investigate
how much more low-frequency there actually is (or might be).
Given the very good coherence of our reconstruction with Trouet's and with the Klesse et al.
reconstructions also on the long-term fluctuations, we did not perform further tests using different
standardization approaches on raw data. The RCS approach with MXD series was also used in a recent
publication of Büntgen et al., 2017, J. Climate). Our paper already presents several different approaches
(for climate sensitivity analysis, HSTC construction and two approaches to temperature reconstruction -
regression and scaling; RDA analysis), we think that this additional analysis would overload the reader.
Moreover, in our opinion, the RCS approach is better performing than splines or the classic negative
exponentials (previous Refereee2's comments) for preserving low-frequency signals, especially with
TRW that present larger widths in the 'young' period of the individual series. By using the same
standardization approach both for MXD and TRW data we avoid the possible introduction of different
frequency responses that would then impede all the comparisons between MXD and TRW in the
analyses of 'Climate sensitivity' and 'Climate sensitivity through time'.
Additionally, I am still extremely cautious of the application of RCS on TRW at sites with n<10-15. If
you use a "10% spline" (which in your case comes close to 15-20 years with the "younger" broadleaf
samples) to build the RC, the RC is potentially very noisy (or wiggly). And if your 3-9 samples have a
narrow age range you essentially take out most of the low frequencies you intended to retain and your
RC at higher ages is probably more flexible at higher ages (due to only very few samples) than the stiff
tail of a negative exponential curve.
At each site the individual series passed a rather robust selection: one of the fixed criteria was "ii) the
individual series correlation with the respective site chronology had r > 0.3" (p. 6 l. 14; Table 2). This
criterion, together with the minimum age of 100 yr length for each series, let us be rather confident that
the resulting Regional Curve used for indexing the raw series is also representative of the growth trends
at each site (please note that for the construction of a RC, all series are aligned to tree age, therefore the
portions with lower sample replication due to the different series lengths are the older ones, where tree-
growth is usually more stabilized).
At sites presenting so few (3-9), albeit well correlating, individual series, we only took the resulting
indexed individual series for constructing the HSTC chronologies (how many series from what site
finally entered in the HSTC used for the climate reconstruction could be further investigated). Of course
having more series at all sites would be better, however with our approach no biases were introduced in
the subsequent analyses. Actually, within all sites and parameters only two TRW chronologies
presenting less than 15 trees were used (namely the ITRDBITAL017 14 trees, and the ITRDBITAL008
12 trees; Table 2), whereas for MXD only one chronology presenting less than 15 trees was used
(ITRDBITAL008 12 trees). We underline again that no site chronology constructed with the RCS
method was based on less than 12 trees.

Not giving an actual number, Esper et al. 2003 and Briffa & Melvin 2011 propose "the more samples
the better", which between the lines is a minimum replication per year at 10 but coming from a
population of >30 in total. Specifically Melvin (2004, Historical Growth Rates and Changing Climatic
Sensitivity of Boreal Conifers, Section 6.3.3), stated you actually would need 62 samples per year for
RCS to get the same per year standard deviation and confidence intervals as a 30-year spline
chronology with n=10 (using Torneträsk and Finish-Lapland chronologies). "The cost for the inclusion
of lowfrequency variance is a requirement for greater tree replication in order to maintain similar
confidence levels."
More samples is better for getting closer to the population mean (TRW o MXD), and for stabilizing the
useful statistics used for assessing the chronology quality in relation to sample replication and
variability. However having more samples does not mean a better climatic signal in the chronology, as it
is also evident in Leonelli et al., 2016 Climatic Change (DOI 10.1007/s10584-016-1658-5), Fig. 4:

[Figure]

**Fig. 4** Site fitness (SF) expressed as the percentage of HSTT series with respect to the total of series available at each site (*red line*). Mean values, standard deviation, median, maximum and minimum values of SF are reported in the included table

The site fitness (SF) index, expressed as the percentage of HSTT (highly sensitive to temperature) series
with respect to the total of series available at each site, is sometimes very low even at sites presenting
more than 100 individual series.
For example, at ITRDB SWIT219 site (https://www.ncdc.noaa.gov/paleo/study/12790), code 26 in the
above figure) with a total of 123 available series, the SF index barely reached the value of 10%.
Although somewhat arbitrary it is common practice to set the EPS threshold to 0.85. The inclusion of
EPS values down to 0.7 in your study tells me a lot about the "weak" coherence within your RCS
chronologies (even the ones with "higher" replication of 16) during the common 1880-1980 interval. I
assume the statistics would be higher (=more robust chronology) if you used a stiffer spline (~150
years) or negexp detrending instead. What are the statistics for the final RCS-HSTC-chronologies, are
they >0.85?
We would have liked higher EPS values for our sites, however this statistics is not the only way for
assessing the chronology quality. Chronologies with low EPS were more frequent in the TRW than in
the MXD where only 3 sites over the 8 used, presented an EPS<0.8 This statistics has been added also
for the HSTC in the revised version of the ms. (Fig. 6 caption).
**Editor Decision: Reconsider after major revisions** (16 May 2017) by Jürg Luterbacher
Comments to the Author:
dear authors
we have now received two Reviews, reviewer is rather positive but has some important Points to be
addressed. Reviewer 2 is more critical in many aspects of the paper. I tend to reject the paper as your
preliminary answers to the reviewers request are not fully convincing but give you the chance for revisions. Therefore I ask you to address all the suggestions/corrections by the reviewers and send a revised version of the paper. It will then go again in review, the final decision will be made at that stage. As an additional point to address from the Editor is to compare your reconstruction with recent gridded temperature reconstructions based on instrumental data taking the closest gridpoints and compare with your reconstruction.

The comparison between tree-ring based temperature was put in the new Fig. 6C and in Table 4. We put the comparisons between JJA temperature and the tree-ring based reconstructions of AS (North Eastern Mediterranean and central and southern Italy) and JAS (Greece) temperature in the Online Material 1, being the comparisons between different variables and between different dataset.

We did not use the spring and autumn gridded dataset of Xoplaki et al. (2005) and the precipitation gridded dataset of Pauling et al. (2006), as our dataset of AS, is more linked to summer temperature.

The comparison between tree-ring based temperature is now performed in Table 4 and in the new version of Fig. 6C that was also enhanced with information on past volcanic eruptions (Referee#1) and mean hemispheric values of incoming solar radiation. The comparisons between Luterbacher et al. (2004) JJA temperature and the tree-ring based reconstructions of AS (North Eastern Mediterranean and central and southern Italy) and JAS (Greece) temperature is presented in the Online Material 1, as it concerns different variables and different datasets. It is performed considering also JJA, JAS and As temperature records from Brunetti et al. (improved version of Brunetti et al., 2006 dataset), whereas we do not use the spring and autumn gridded dataset of Xoplaki et al. (2005) and the precipitation gridded dataset of Pauling et al. (2006), as our dataset of AS, is more linked to summer temperature.

We do not discuss the differences between the Luterbacher et al. and the Brunetti et al records (both based on instrumental data) as this issue is outside the goals of this paper. We underline however that the Brunetti et al. records are much more appropriate for the area covered by our chronologies and that they have been homogenized taking into account the warm-bias problem which seems to affect many early instrumental temperature series (see Böhm et al., 2010, Climatic Change, 101, 41–67).

The Authors, 26 June 2017

[revised manuscript text omitted]

---

## Author Response (AR2)

Dear Prof. Luterbacher,

we have responded to the points raised by the Reviewer #2, and hereafter you can find all the details. After treating the data as suggested by the Reviewer, the quality of the reconstruction has increased and some figures were redrawn accordingly. We also added a correlation map with precipitation (Fig. 7), as tree-ring MXD from the study region holds also a strong signal of drought conditions and precipitation (Fig. 3).

We hope that now the ms. has reached the quality standard of CP.

Kind regards,
Giovanni Leonelli

Reviewer#2 comments - received 17 July 2017
Our responses in BLUE

The authors addressed and clarified many of my initial comments in a sufficient manner, but still a couple of issues persist, where the authors fall short to make the best out of their analysis, results and discussion.
Thank you for giving us this further opportunity to improve the ms.

**Major comments:**

P6 L36: Why didn't you use the more commonly used bi-weight-robust mean to average your chronologies? There are two series in ital015x (around the year 1700) and one of the long series (620012) in ital012x that have extremely low values over a prolonged period of time, dragging your chronology values down heavily (I presume they are left in the HSTC approach). That is usually no big issue if you have a high sample replication. But in your case those 3 series have a huge impact on your reconstruction when your replication is only around 10 or lower. Using bi-weight over arithmetic mean buffers significantly against those outliers. Your earliest 50 years would actually be quite a bit closer to Trouet and Klesse (you "greyed-out" your chronology before 1710, I know).
We tested several combinations of HSTC MXD chronologies constructed following the suggestions given in this and the following comment. The best combination in terms of model stability for temperature reconstruction also with the scaling method was: normalize the indexed series; apply an arithmetic mean; perform average stabilization. All the other combinations (also including the biweight mean), resulted in less stable models for temperature reconstruction and were therefore discarded.
We found several authors simply applying an arithmetic mean in the construction of the mean chronology (e.g., with MXD series, Klusek et al., 2015, Dendrochronologia; with multi parameters series, Esper et al., 2006, Trace), probably this is the best way with the HTSC approach.

Portion of the chronology with low sample replication ( < 10).
In the early portion of the chronology, one (not 3) indexed series shows low MXD values that indeed may influence the mean; this is particularly evident especially after normalizing the series (see series #630071 in the following figure). The series #620012 plus probably other two series mentioned by the Referee are not part of the HSTC group used for the construction of the MXD chronology.

[Figure]

*Indexed normalized MXD series considered for the HSTC chronology. Yellow line, the series #630071, itrdbital015, a P. leucodermis sample); green line the arithmetic mean; the vertical yellow line demarks the year 1713.*

In order to improve the HSTC approach over the early period with the chronology showing an EPS < ~0.8 (i.e. before 1713 in the first version of the HSTC chronology, the greyed-out portion), we considered the yearly difference of the indexed normalized series from the mean and discarded the early portion of the series exceeding 2.5 standard deviations in a given year. The series exceeding this threshold is only the #630071 since 1713 included, whereas all the other fall within a common variability. Therefore we discarded the values of the #630071 series for the period 1692-1713, we re-normalized it and we recalculated the final version of the HSTC chronology used for the temperature reconstruction.

The issue related to the influence of outliers in the arithmetic mean computation in presence of low sample replication was described above. We posed a threshold to acceptable variability in the indexed series: all indexed series (one) exceeding 2.5 standard deviations from the mean were truncated. As alternative option, we could simply discard the whole portion of chronology showing an EPS < ~0.8.

The resulting final HSTC chronology put in the ms. has now an EPS > 0.791 since 1714 (thus we greyed-out and discarded the early portion before 1714 in Fig. 6). From 1714 to 1980 the chronology has a sample replication >= 10 and a good signal stability with an EPS > 0.85 since 1735.

| year | # of samples | EPS |
|------|--------------|-----|
| ... | | |
| 1710 | 7 | 0.726 |
| 1711 | 8 | 0.752 |
| 1712 | 8 | 0.752 |
| 1713 | 9 | 0.773 |
| 1714 | 10 | 0.791 |
| .... | | |

Furthermore, by simply averaging across all sites, you don't account for systematic differences between sites, when samples and sites drop out of your reconstruction. Probably more robust would be to normalize all individual series over the common period (e.g. 1880-1980) and then apply the bi-weight mean. This way the chronology is much more buffered against sudden site/series/species replication changes. It definitely affected the long-term slope of the chronology when I use all 7 sites.

The indexed normalized series are rather synchronous and present similar MXD values: this probably explains why applying a single arithmetic mean is a valid approach. Furthermore, having discarded the problematic portions in the early period with low sample replication and low EPS, we have now ameliorated also the 'greyed-out' early portion of the reconstruction showing less than 10 samples per year (this portion, however, lies in the period already discarded for the low EPS value in the chronology).

All indexed series are now normalized before averaging, as reported above.

Additionally, and equally - if not more - important for constructing the chronology for climate reconstruction purposes, I do not see you using variance stabilization, which should be a standard step producing a valid reconstruction. Using all of the 7 ITRDB MXD sites and using the arithmetic mean without variance stabilization, the 30-year running standard deviation increases from ~0.03 to ~0.05 from 1870 back to 1750, simply due to the decrease in replication (from close to 100 to only 30 samples). I'm pretty sure this effect appears similarly with only the HSTC series (~half of the dataset I tested) and the negative CE using the scaling approach could be a result of this.

In the construction of the HSTC chronology used for the temperature reconstruction, we now applied an arithmetic mean to average the indexed series after normalizing them over the resulting common period 1879-1962. We also applied a variance stabilization of the chronology, following Osborn et al., 1997.

The obtained HSTC MXD chronology (blue line in the figure hereafter) shows an EPS > ~0.8 since 1714 (EPS = 0.791; 10 trees) and >0.85 since 1734 (15 trees). The blue line depicts the new chronology used for temperature reconstruction, the red line depicts the chronology used in the previous version of the ms.

[Figure]

*Blue line, the new MXD HSTC chronology (corrected in the early period with low sample replication, normalized indexed series and variance stabilization); red line the previous version of the HSTC chronology. Bottom line: sample replication (secondary y axis).*

The reconstruction is now satisfying in both the one-century-long (inverted) calibration-verification periods and it is stable over time, with CE values showing always positive values (see new Table 3 in the ms). We also tested the possibility of using a biweight mean to average the indexed and normalized series, but the temperature reconstruction showed a higher instability of the model (with CE showing slightly negative values over the verification period 1781-1880: -0.078).

As an overall consideration we are more concerned of anomalous fluctuations in the most recent periods like those around 1913 and especially 1977 as tracked, e.g., by the Trouet's (2014) reconstruction, that underline late summers nearly as cold as in the 1810s (i.e., Little Ice Age peak period), than in the early period, with the chronology showing lower sample replication and lower EPS (Fig. 6C).

P10 L25-27: It might be better to say: "Additionally to the stronger AS temp influence on MXD in the northern chronologies the effect of summer precipitation/drought becomes equally stronger at the southern sites." I wouldn't stress the elevation influence here in the second sentence, because I assume the main effect is actually latitude/longitude, that causes your gradient in average precipitation. You simply don't have the trees at high/low elevations in northern/southern locations. With elevation, temperature should decrease, and precip increase. The increasing drought correlation with increasing elevation is likely the result of the "unbalanced" tree-ring network.

Ok. We modified the text.

Did you check what the pure ABAL MXD HSTC chronology would look like? Isn't your analysis suggesting a cleaner and purer temperature signal there? Makes all sense to me, that when the average precip levels go up, the sensitivity towards precip/drought variation decreases. Again, KNMI (using only ABAL samples) tells me it would improve!

So, I'm really curious how the correlation stats would change, maybe even increase, if you created an ABAL-only chronology? I find that diving a little more into this matter would strongly increase the value of this paper, together with an outlook statement that thanks to the RDA you find (that although it might be easier to find old and still standing dead PILE trees on/around Monte Pollino – ok RDA doesn't tell you that but you get what I'm aiming at I hope), the MXD signal there is potentially "confounded" by drought. Updating ABAL MXD + finding more old ABAL trees and extending present data with building material (? Not familiar with dendroarcheology in central Italy, but I guess there should be houses built with ABAL?) in central Italy seems a promising line of research for the coming years. That's in my opinion the most exciting message I get from your paper. It deserves more room in the discussion and would certainly contribute the most to the advance of dendro-paleo-climatology in central Italy. P11 L28-32 is a bit thin in this respect.

One of the purposes of this paper is to reconstruct a temperature signal valid for the whole Italian Peninsula. In this view, removing *P.leucodermis* specimens would further drop the sample replication in the early period, thus likely resulting in higher instability of the signal in the chronology and a shorter period of temperature reconstruction. Future research could take in account what the Referee suggests here above, hopefully also with the effort of more research groups.

1     Given the change in the reconstructed temperature series, we re-computed also the spatial correlations.
2     Unfortunately the CRU TS/E-OBS 13.1 dataset was not available anymore in the KNMI, and with
3     the newly available version CRU TS/E-OBS 15.0, we noticed locally some unexpected responses
4     that were not present in the 13.1 version and that are likely due to problems in the climatic dataset.

[Figure]

[Figure]

8 *Spatial correlation of the MXD HSCT and the CRU TS/E-OBS 15.0: r values (top) and their statistical*
9     *significance (bottom).*
10
11
12 We therefore preferred to use another dataset, namely the CRU TS 4.0 and compared our AS
13     temperature reconstruction with both AS mean temperature and AS mean precipitation (see new
14     Fig. 7).
15
16 We also updated Table 4 and all the analysis performed in online materials.

P10 ~L37-41: Is the climate signal in RW also instable if both tree-ring chronologies and climate data are high-pass filtered? I already raised that topic in the first round and the authors actually responded they would discuss that in the new ms (*"deepening on correlation coefficient trends, as suggested by Refereee2, will be added in the revised version of the ms"*). I really would like to see this additional analysis here, as you don't discuss the detrending issue at all in this section. Because you use RCS with a dataset that is not really fit for it, one could assume that there might be some artificial low-frequency trends in the HSTC RW chronology (due to site and sample replication changes that your simple averaging approach simply cannot deal with) that decrease your climate correlation before 1900 in the conifers. Or is it also the high-frequency signal that is weaker to non-existent (due to early "bad" climate data, few records, higher uncertainty of interpolation of climate data, or because of points you did discuss)? Please discuss this!

The detrending approach used at the site level is common to many other researches already published, however there are no other possible ways if high-quality data availability is scarce in a given site. We remind that at each site, we selected only highly correlating series showing a common signal, in order to avoid large amount of samples bringing different signals than climatic ones (see Methods — *Site chronologies*). Averaging the indexed series using an arithmetic mean is also common (see references here above reported), and probably the best approach for the HSTC, given the good results obtained in the reconstruction.

'Early "bad" climate data' issue: for what concerns the climate series, they are based on

35 thermometric stations with data before 1900, of which 16 are below 44° N of latitude.
17 thermometric stations with data before 1870, of which 5 below 44° N of latitude.
8 thermometric stations with data before 1850, of which 1 below 44° N of latitude.

For precipitation, data availability is much larger.

As suggested by the Referee, we high-pass filtered the series (with a 20 yr gaussian, σ = 4 yr), and obtained the following correlations, showing no great changes between original and high-pass filtered series.

| HSTC code | Climate variable | Correlation coefficient | Period |
|---|---|---|---|
| **1** | AS temp | 0.53 | 1774-1980 |
| high-pass filtered series | | 0.54 | |
| **1 normalized series\*** | AS temp | 0.66 | 1774-1980 |
| high-pass filtered series | | 0.61 | |
| **2** | JJA precip sum | -0.56 | 1811-1980 |
| high-pass filtered series | | -0.53 | |
| **3** | JJAS SPI | -0.55 | 1811-1980 |
| high-pass filtered series | | -0.57 | |
| **4** | A-1 temp | -0.23 | 1775-1987 |
| high-pass filtered series | | -0.3 | |
| **5** | JJ precip sum | 0.39 | 1857-2013 |
| high-pass filtered series | | 0.42 | |
| **6** | JJA SPI | 0.36 | 1857-2013 |
| high-pass filtered series | | 0.4 | |

*\* this is the MXD HSTC series used for the AS temperature reconstruction.*

Following the Referee suggestion, we also performed moving interval analysis on the HSTC chronologies "1", "1 normalized series" (the HSTC chronology used for temperature reconstruction), and the "4" (i.e. the chronology showing the greatest changes in significance of the correlation coefficients).

[Figure]

*Bootstrapped moving correlation analysis with a 60 yr time window, performed over the maximum period available for the **high-pass filtered** HSTC chronologies "1", "1 normalized", "4" and their respective **high-pass filtered** climate variables (AS temperature, AS temperature and A-1 temperature, respectively). Bold lines depict statistically significant values (p<0.05) of r.*

No great changes can be noticed between r patterns in the original series and the high-pass filtered ones (see Fig. 5 for comparison). Conifer MXD ("1" and "1 normalized") are always significantly correlated with AS temperature over the whole period of analysis, with a slight fluctuation towards

higher r values in the late periods, although with lower absolute values. A slightly more stable pattern in correlation coefficients around 0.6 is noticed for the "1 normalized" (i.e. the MXD HSTC used for temperature reconstruction) with respect to the "1", but the patterns are the same.

Conifer RW ("4") passes from non-significant values in the early period to significant ones, even though with high pass filtered series. This pattern is the same as in Fig. 5, although with high-pass filtered series, the passage to significant values happens since the time windows ending in 1885 and not in 1900.

As already underlined, the temporal stability in climate correlations was tested on HSTC chronologies of RW (broadleaf and conifer) and MXD (conifer), for evaluating the reconstruction potentials and the possible biases in past climate reconstructions. As we found the same patterns of r in high-pass filtered series, we are confident that the approach here used is solid.

Furthermore: Is the conifer RW (lag1) and conifer MXD actually negatively correlated? How about conifer MXD and broadleaf RW? Could there be a way of disentangling drought effects on MXD in the future using drought sensitive RW from different species? I'm aware of different auto-correlative structures of the two parameters, but discussing or hypothesizing about these issues could be an interesting addition to the discussion section.

The Referee asks to compare two different objects, as the two HSTC chronologies "1" and "4" derive from trees located in different site over Italy, that are sensitive to AS temperature (chronology 1) and A-1 temperature (chronology 4), respectively. Probably the relation the Referee is referring to is valid for same trees from the same sites.

Just to be sure, we also checked for possible correlation and obtained the following results:

| HSTC code | lag 0 | mxd lagged -1 | mxd shifted +1 |
|---|---|---|---|
| "1" vs. "4" | 0.04 | 0.14 | -0.11 |
| "1 normalized series" vs. "4" | 0.05 | 0.16 | -0.09 |

Forgive me if I constantly keep over-looking it, but what is the final temperature target? You do not specify exactly the region over which you averaged the climate in the methods. Or is it simply the average of the site-corrected climate data? At the end of 2.2 this description is missing.

The MXD-based reconstruction is performed using the driving climate variable (DCV) 'AS temperature' (p. 6, l. 23 and following) constructed from the different AS temperature series at the various MXD sites showing significant correlation in the site-level analysis on site chronologies (Fig. 3). The target is therefore a mean August September temperature for the whole Peninsula, since MXD sites are distributed in the whole Peninsula. In particular, the regional AS temperature series used for the reconstruction was build starting from the climate series specific of the following sites (i.e. the sites showing significant correlations between reference chronologies and August and September temperature):

ITRDBITAL008
ITRDBITAL009
ITRDBITAL010
ITRDBITAL011
ITRDBITAL012
ITRDBITAL015

The climatic data from the following sites (of *P. nigra*) were not included in the construction of the regional climatic series, as we found no significant correlation between site chronologies and August or September temperature:

ITRDBITAL013
ITRDBITAL016

We modified some parts of the paragraph.

And what's the $R^2$ for the full period? You only report it for the split periods.
We added the $R^2$ for the full period in Table 3

Figure update: Figure 4B was updated, as some series were wrongly assigned in the computation of the Site Fitness.

**Minor comments:**

P4 L17: This is a rather abrupt change to describe the purpose of your study. I would give it one or two more sentences. Something along the lines of: "As separate climate (temperature) reconstruction for Italy has been published to date the goal of this study was to screen the ITRDB for suitable data. we make use of the ITRDB to investigate RW and MXD climate signals across Italy. After screening … -> temperature reconstruction …
Our main objectives are: i) … ii) …"
Ok. thank you.

P6 L16: "variables" instead of "variable"
Ok.

P6 L21: delete [in presenting HSTC trees], simply say "… we calculated the Site Fitness, representative of the percentage of selected HSTC series of conifer MXD with respect to …"
Ok

P8: Please consider revising the first 8 lines. There are some unnecessary repetitions.
Something along the lines: "We find that the strength of the AS signal correlates positively with latitude

(mean precip) and negatively with elevation (longitude).. The RDA analysis reveals that both parameters are on opposing sides of the first two axes explaining NNN% of the variance of the dataset. … "

Ok

P8 L9: Concerning the site fitness – instead of "for what concerns"

Ok

P8 L38-40: Something wrong in that sentence, maybe easier something like: "… the MXD reconstruction matches very well the temperature variability in Italy south of the Po plane and the western Balkan area (r>0.6). Correlations above (e.g.) 0.4 are still found throughout the Alpine arc, the central Balkan, as well as Tunisia."

Ok

P9 L4-5: do you mean: strong signal in MXD independent of species?

Yes

P10 L4: Pinatubo did not erupt in 1914! Do you maybe mean Novarupta 1912? There's no VEI>=6 in 1914, to my knowledge (and your figure 6).

Yes, of course... Ms. modified accordingly.

P10 L26/27: no commas after MXD

Ok

P10 L26: P. leucodermis

Ok

P10 L36: concerning the temperature signal

Ok

Figure 6: Mount Pinatubo is not on your axis, as your graph ends in 1980.

Yes

Figure 7: Is the E-OBS/CRU TS a spliced product? E-OBS usually starts only in 1950, I can't find a clear description on KNMI what this actually is. The correlation picture (RCS with all 7 sites and 160 MXD series) looks almost the same compared to using CRU TS4.0, which makes me wonder, what exactly the benefit of the HSTC approach is, since you shorten your potentially "reliable" chronology by 80 years (using all series you would have an EPS>0.85 back to 1650).

CRU TS/E-OBS is a combination of datasets. In the current version, given the problems raised with CRU TS/E-OBS v 15.0, we now choose the CRU TS 4.0. The HSTC approach is a tree-based approach that has the advantage of focusing the reconstruction only on trees whose sensitivity to climate has been checked over centuries.

The Authors, 25 August 2017

[revised manuscript text omitted]